# A realistic physical model of the Gibraltar Strait

Axel Tassigny<sup>1,★</sup>, Stef L. Bardoel<sup>1</sup>, Thomas Valran<sup>1</sup>, Samuel Viboud<sup>1</sup>, Louis Gostiaux<sup>2</sup>, Joël Sommeria<sup>1</sup>, Lucie Bordois<sup>3</sup>, Xavier Carton<sup>4</sup>, and Maria Eletta Negretti<sup>1,★</sup>

**Correspondence:** Maria Eletta Negretti (eletta.negretti@legi.cnrs.fr)

**Abstract.** We present a large-scale laboratory model of the Strait of Gibraltar that reproduces realistic topography, tidal forcing, stratification, and rotation, enabling controlled investigation of key exchange processes linking the Mediterranean Sea and Atlantic Ocean. Velocity and density measurements confirm dynamic similarity with ocean observations. Analysis of the flow near Camarinal Sill shows that bottom boundary layers are the primary source of turbulent kinetic energy, exceeding contributions from shear at the interface between Atlantic and Mediterranean waters. The enhanced role of bottom-generated turbulence is linked to separation of the Mediterranean gravity current induced by an adverse pressure gradient during outflow, providing a new explanation for the well-documented detachment of the Mediterranean plume west of the sill. This detachment intensifies during spring tides, driving diluted waters farther into the Atlantic, while during neap tides bottom-generated and interfacial turbulence coincide, offering a consistent explanation for the high dissipation rates reported in field measurements. Overall, tidal forcing promotes full-depth mixing, with up to 20% density reduction west of CS and oscillatory 20% variations east, consistent with field data, and simultaneously introducing an important phase shift between velocity and density fields, with implications for parameterizing turbulent exchange and definition of the composite internal Froude number for reliable diagnose of hydraulic control. During spring tides, hydraulic control is intermittently lost during inflow and this loss propagates eastward, while additional control points arise west of the sill. Neap tides exhibit signatures of control which persist much longer during a tidal cycle as compared to spring tides, but does not propagate to the east when the tide reverses. Transport and energy budgets reveal strong longitudinal and transverse variability, highlighting the need for fully three-dimensional diagnostics. Volume transport, dominated by transverse topographic variability, exceeds salt transport by two orders of magnitude, confirming net Atlantic inflow. A phase-lagged internal bore release between the northern and southern transects is observed, consistent with field observations, and we show that it is independent of barotropic effects or Kelvin waves. These results demonstrate that high-fidelity laboratory modeling can capture the essential three-dimensional dynamics of energetic straits and provides a powerful complement to observational and numerical approaches.

<sup>&</sup>lt;sup>1</sup>LEGI UMR5519, Univ. Grenoble Alpes, CNRS, Grenoble INP, Grenoble, 38000, France

<sup>&</sup>lt;sup>2</sup>CNRS, Ecole Centrale de Lyon, INSA Lyon, Universite Claude Bernard Lyon 1, LMFA, UMR5509, 69130, Ecully, France

<sup>&</sup>lt;sup>3</sup>Service Hydrographique et Océanographique de la Marine (SHOM), Brest, France

<sup>&</sup>lt;sup>4</sup>University Brest, CNRS, Ifremer, IRD, Laboratoire d'Océanographie Physique et Spatiale (LOPS), IUEM, Plouzané, France ★These authors contributed equally to this work.

#### 1 Introduction

Density-driven flows interacting with topography generate dense currents, or gravity currents, which play a crucial role in transporting water masses, heat, and momentum in the oceans. These flows exemplify mesoscale dynamics that give rise to small-scale processes, including boundary layers, strong shears, instabilities, and kilometer-scale sub-mesoscale eddies. The mixing induced by these processes affects the stabilization depth of water masses and ultimately influences the global thermohaline circulation (Price and O'Neil Baringer, 1994; Danabasoglu et al., 2010). Oceanic examples include the Denmark Strait (Käse et al., 2003), Arctic and Antarctic continental shelves (Aagaard et al., 1981; Muench et al., 2009), and marginal seas where dense waters form due to high evaporation, such as the Red Sea, the Arabian Gulf, and the Mediterranean Sea (Peters and Johns, 2005; Vic et al., 2016; Baringer and Price, 1997).

The Strait of Gibraltar connects the Mediterranean Sea to the Atlantic Ocean through a narrow passage between southern Spain and northern Morocco. It is both a major maritime route and the Mediterranean's only outlet to the global ocean, making it one of the most extensively studied regions in oceanography (Armi and Farmer, 1988; Baringer and Price, 1999; García-Lafuente et al., 2007, 2013). Beyond its strategic importance, it supports significant biological productivity (Echevarria et al., 2002), influencing fisheries and the regional economy. Oceanographically, the Mediterranean Outflow Water strengthens the Atlantic Meridional Overturning Circulation (AMOC) (Reid, 1979), stabilizes North Atlantic climate within natural variability over the past 2 Ma (Rogerson et al., 2012), and contributes to the Azores Current and the Gulf of Cadiz Current systems (Jia, 2000; Peliz et al., 2007). A deeper understanding of the Strait's dynamics is therefore crucial for improving regional climate modeling and predicting the influence of Mediterranean salinity on North Atlantic circulation.

From a fluid dynamics perspective, the Strait provides a clear example of how fine-scale processes influence larger-scale ocean dynamics (Hilt et al., 2020; Roustan et al., 2023). With depths ranging from 175 m to 1000 m and widths of approximately 15 km, the Strait channels an exchange of roughly 0.8, Sv in each direction (Soto-Navarro et al., 2010), supplemented by a net barotropic transport of 0.05, Sv to balance Mediterranean freshwater deficits (Bryden et al., 1994). Strong tidal currents interact with the bathymetry to generate flows exceeding one meter per second, which evolve on hourly timescales. These dynamics create a highly constrained environment in which small-scale mixing processes significantly impact large-scale exchanges.

Most previous studies of the Strait have relied on idealized numerical or experimental models. These have explored internal hydraulics (Farmer and Armi, 2001; Pawlak and Armi, 1996; Zhu and Lawrence, 2000; Fouli and Zhu, 2011), shear instabilities driving turbulent mixing (Baines, 2002; Negretti et al., 2008; Odier et al., 2014), and the effects of Earth's rotation on vorticity, stratification, and turbulence (Negretti et al., 2021; Tassigny et al., 2024; Rétif et al., 2024). While these models provide valuable insights, they often fail to capture the feedback of localized, small-scale turbulence on broader circulation patterns (Danabasoglu et al., 2010; Ferrari and Wunsch, 2009; Ferrari et al., 2016). Accurate representation of non-hydrostatic, multi-scale gravity current dynamics remains a major challenge for numerical simulations, despite advances in two-layer, three-dimensional, and non-hydrostatic modeling approaches (Brandt et al., 1996; Izquierdo et al., 2001; Sannino et al., 2002; Sánchez-Garrido et al., 2011; Sannino et al., 2014; Naranjo et al., 2014). Embedded high-resolution grids and focused modeling of the Strait have improved simulation of Mediterranean stratification and convective events, but uncertainties remain

regarding internal hydraulics, instabilities, energy dissipation, and the generation and propagation of internal waves.

Observational limitations further complicate our understanding. In situ measurements, while increasingly detailed, cannot always capture the full three-dimensional, intermittent nature of the flow, particularly over steep bottom slopes. Remote sensing provides broader spatial coverage but is limited to surface observations and cannot resolve fast-evolving small-scale processes. Laboratory experiments, by contrast, offer a controlled environment to validate numerical models and interpret observational data, allowing detailed study of internal waves, instabilities, hydraulic jumps, and mixing processes (Rubino et al., 2020; Gacic

et al., 2021; Pierini et al., 2022; Shi et al., 2022; Pirro et al., 2024).

In this study, we present the first large-scale physical model of the Strait of Gibraltar, including the Gulf of Cadiz and the westernmost Alboran Sea, achieving an unprecedented level of realism. The model incorporates tidal and baroclinic forcing, rotational effects, realistic bathymetry, and a sufficiently large domain to capture the synoptic interaction of small-scale processes with regional circulation, similar in approach to previous scaled models of the Luzon Strait (Mercier et al., 2013). The experimental configuration reproduces global internal hydraulics, small-scale turbulence, internal wave generation, and Mediterranean Outflow propagation with realistic velocities and dilution patterns. The experiments reveal the critical role of bottom boundary layers and topography in shaping flow dynamics, transport, dilution, and vorticity production, factors previously assumed secondary relative to interface shear or tidal forcing. Analysis of three-dimensional transport and energy budgets highlights strong spatial variability, demonstrating that two-dimensional or averaged fields cannot reliably represent fluxes. This paper focuses on the design and implementation of the physical model (Sections 2–3) and presents results on overall flow dynamics within the Strait (Section 4), followed by conclusions (Section 5). Detailed analyses of internal wave dynamics, mix-

dynamics within the Strait (Section 4), followed by conclusions (Section 5). Detailed analyses of internal wave dynamics, mixing over Camarinal Sill, Espartell, west Espartell, and Mediterranean Outflow propagation in the Gulf of Cadiz are presented in companion papers (Tassigny et al., 2026c, b, a; Bardoel et al., 2026).

#### 2 The physical model

### 2.1 Governing equations

We distinguish the horizontal velocity  $\hat{\mathbf{u}}_{\mathbf{H}}$  and vertical velocity  $\hat{w}$ , as well as temporal  $(\partial_{\hat{t}})$ , vertical  $(\partial_{\hat{z}})$  and horizontal  $(\nabla_H = \mathbf{e}_{\mathbf{x}}\partial_{\hat{x}}+\mathbf{e}_{\mathbf{y}}\partial_{\hat{y}})$  partial derivatives. We consider the equations under the Boussinesq approximation, implying variations of density being small  $(\delta\rho/\rho\ll1)$ . Thus the density is assumed constant (equal to  $\rho_0$ ) for the inertial terms of the momentum equation, but the small density variations are essential for the gravitational forcing. The bottom topography is represented by the variable  $\hat{z}=-h_b(\hat{x},\hat{y})+\eta(\hat{x},\hat{y},\hat{t})$ , where the axis  $\hat{z}$  is directed upward,  $h_b(\hat{x},\hat{y})$  is the depth of the bottom topography with respect to  $\hat{z}=0$ , and the free surface is given by  $\hat{z}=\hat{\eta}(\hat{x},\hat{y},\hat{t})$  assumed at rest at  $\hat{z}=0$  and the pressure  $\hat{p}=0$  at  $\hat{\eta}$ . The buoyancy is given by  $\hat{b}=-g(\rho-\rho_0)/\rho_0$ , where  $\rho_0$  is a reference density assumed to be the one of the Atlantic water. The pressure  $\hat{p}$  is composed of a hydrostatic part  $\hat{p}_0$  and a non-hydrostatic component  $\hat{p}_{\rm nh}$  so that  $\hat{p}=\hat{p}_0+\hat{p}_{\rm nh}$ . The hydrostatic pressure  $\hat{p}_0$  is given by a barotropic and a baroclinic part such that

$$\partial_z \hat{p}_0 = -\rho_0 g + \rho_0 \hat{b}. \tag{1}$$

The governing equations read then

$$\partial_{\hat{t}}\hat{\mathbf{u}}_{\mathbf{H}} + \hat{\mathbf{u}}_{\mathbf{H}} \cdot \nabla_{H}\hat{\mathbf{u}}_{\mathbf{H}} + \hat{w}\partial_{\hat{z}}\hat{\mathbf{u}}_{\mathbf{H}} = -\frac{1}{\rho_{0}}\nabla_{H}(\hat{p}_{0} + \hat{p}_{\mathsf{nh}}) - f(\mathbf{e}_{\mathbf{z}} \times \hat{\mathbf{u}})_{H} + \nu\left(\nabla_{H}^{2}\hat{\mathbf{u}}_{\mathbf{H}} + \partial_{\hat{z}^{2}}\hat{\mathbf{u}}_{\mathbf{H}}\right)$$
(2a)

$$\partial_{\hat{x}}\hat{w} + \hat{\mathbf{u}}_{\mathbf{H}} \cdot \nabla_{H}\hat{w} + \hat{w} \cdot \partial_{\hat{z}}\hat{w} = -\partial_{\hat{z}}(\hat{p}_{\mathsf{nh}}/\rho_{0}) + \nu \left(\nabla_{H}^{2}\hat{w} + \partial_{\hat{z}^{2}}\hat{w}\right) \tag{2b}$$

$$\nabla_{\mathbf{H}} \cdot \hat{\mathbf{u}}_{\mathbf{H}} + \partial_z \hat{w} = 0 \tag{2c}$$

$$\partial_{\hat{r}}\hat{b} + \hat{\mathbf{u}}_{\mathbf{H}} \cdot \nabla_{H}\hat{b} + \hat{w}\partial_{\hat{z}}\hat{b} = 0$$
 (2d)

$$\partial_{\hat{r}}\hat{\eta} + \hat{\mathbf{u}}_{\mathbf{H}} \cdot \nabla_{H}\hat{\eta} = \hat{w} \text{ at } \hat{z} = \hat{\eta}$$
 (2e)

where  $\nu$  is the kinematic viscosity of the fluid and the scalar diffusivity is neglected.

Let us introduce the non-dimensional variables denoted without hat  $\hat{\cdot}$ :

$$(\hat{x}, \hat{y}) = L(x, y)$$
 
$$\hat{z} = Hz,$$
 
$$\hat{u}_H = Uu_H,$$
 
$$\hat{w} = Ww,$$
 
$$\hat{b} = g'b,$$
 
$$105 \qquad \hat{p}_0 = \rho_0 U^2 p_0,$$
 
$$\hat{p}_{\mathrm{nh}} = \rho_0 \left(\frac{H}{L}\right)^2 U^2 p_{\mathrm{nh}},$$
 
$$\hat{\eta} = H\eta,$$

 $\hat{t} = T_{tide}t$ 

where  $g'=g(\rho_M-\rho_0)/\rho_0$  is the constant reference value of reduced gravity obtained by the initial density difference between the Mediterranean  $(\rho_M)$  and the Atlantic  $(\rho_0)$  densities. The non-dimensional equations are then given by

$$\nabla_H \cdot \mathbf{u}_H + \frac{\partial w}{\partial z} = 0 \tag{3a}$$

$$\partial_z p_0 = -\frac{1}{Fr_0^2} + \frac{b}{Fr^2} \tag{3b}$$

115 
$$St\partial_t \mathbf{u}_H + \mathbf{u}_H \cdot \nabla_H \mathbf{u}_H + w\partial_z \mathbf{u}_H = -\nabla_H (p_0 + \frac{H^2}{L^2} p_{\text{nh}}) - \frac{1}{Ro} (\mathbf{e}_{\mathbf{z}} \times \mathbf{u})_H + \frac{1}{Re} \left( \frac{H}{L} \nabla_H^2 \mathbf{u}_H + \frac{L}{H} \partial_{zz} \mathbf{u}_H \right)$$
 (3c)

$$\frac{H^2}{L^2}\left(St\partial_t w + \mathbf{u}_H \cdot \nabla_H w + w\partial_z w\right) = -\frac{H^2}{L^2}\partial_z p_{\mathsf{nh}} + \frac{1}{Re}\frac{H^2}{L^2}\left(\frac{H}{L}\nabla_H^2 w + \frac{L}{H}\partial_{zz}w\right) \tag{3d}$$

$$St\partial_t b + \mathbf{u}_H \cdot \nabla_H b + w \partial_z b = 0 \tag{3e}$$

120 
$$St\partial_t \eta + \mathbf{u}_H \cdot \nabla_H \eta = w \text{ at } z = \eta$$
 (3f)

where W = UH/L from the continuity equation,  $Fr_0 = U/\sqrt{gH}$  and  $Fr = U/\sqrt{g'H}$  are the external and internal Froude numbers,  $St = L/(UT_{tide})$ , Ro = U/(fL),  $Re = UH/\nu$ . In the vertical momentum equation (3c) we used the hydrostatic pressure decomposition (3b), in which both the external and internal Froude numbers appear, whereas the non-hydrostatic pressure term scales as the squared aspect ratio. From equation (3b), the ratio of the two terms with the external and internal Froude numbers scale like g/g'. Hence, achieving similarity of both  $Fr_0$  and Fr would prevent rescaling of the density. However, since  $Fr_0 \gg Fr$ , i.e. free surface waves are much faster than internal dynamics, the term containing the external Froude number can be neglected.

### 2.2 Scaling

135

To design the experimental model, we first set the maximum available length which determines the ratio of the horizontal scales L. Then we set the vertical scale sufficiently large to minimise viscous and surface tension effects, but not too large to avoid excessive slopes. These considerations lead to the scale factor 25,000 in the horizontal and 2,500 in the vertical (see table 1). This increases the slopes s by a factor 10, which is not critical as long as they satisfy s 

| Parameters                    | ocean                           | experiments             | ratio             |
|-------------------------------|---------------------------------|-------------------------|-------------------|
| Primary                       |                                 |                         |                   |
| L                             | 100 km                          | 4 m                     | 25,000            |
| H                             | 1,000 m                         | 0.4 m                   | 2,500             |
| $H_s(\mathrm{sill})$          | 100 m                           | 0.04 m                  | 2,500             |
| g'                            | $0.03~{\rm ms}^{-2}$            | $0.19~{\rm ms}^{-2}$    | 1/6.3             |
| f                             | $0.8 \ 10^{-4} \mathrm{s}^{-1}$ | $0.1 \ \mathrm{s}^{-1}$ | 1/1,258           |
| $T_{ m tide}$                 | 44,712 s                        | 35.77 s                 | 1,258             |
| Derived                       |                                 |                         |                   |
| $U_g = \sqrt{g' H_s}$         | 1.7 m/s                         | 8.7 cm/s                | 20                |
| $U_b$                         | 1 m/s                           | 5 cm/s                  | 20                |
| $T = L/U_g$                   | $57.7 \ 10^3 s$                 | 46 s                    | 1,258             |
| $R_D = \sqrt{g'H_s}/f$        | 21.6 km                         | 87 cm                   | 25,000            |
| Non-dimensional               |                                 |                         |                   |
| H/L                           | 0.01                            | 0.1                     | 0.1               |
| $Fr = U_g/\sqrt{g'H_s}$       | 1                               | 1                       | 1                 |
| $Ro = U_g/(fL)$               | 0.22                            | 0.22                    | 1                 |
| $Bu = (R_D/L)^2$              | 0.05                            | 0.05                    | 1                 |
| $\mathrm{Re} = U_g H_s / \nu$ | 1.7 10 <sup>8</sup>             | 3.5 10 <sup>3</sup>     | 5 10 <sup>4</sup> |
| $St = L/(UT_{tide})$          | 1.3                             | 1.3                     | 1                 |
| $A_{tide} = U_b/U_g$          | 1                               | 1                       | 1                 |

Table 1. Primary, derived and relevant non-dimensional parameters in the laboratory model of the Gibraltar Strait.

water depth at CS. This estimate gives  $U_g \simeq 1.7\,\mathrm{ms}^{-1}$ . A more precise estimate is obtained by introducing the control condition for the composite Froude number  $G^2 = Fr_1^2 + Fr_2^2 = 1$ , with  $Fr_{1,2}^2$  being the internal Froude numbers of each layer. The velocity is then reduced by a factor  $\sqrt{2}$ , in closer agreement with observation values of 1 m/s. In the experiments, the corresponding height is reduced by a factor 2,500, so the baroclinic velocity is reduced by a factor  $\sqrt{2,500} = 50$  if we keep the same relative density difference. To increase the velocity, hence the Reynolds number, we choose a higher density difference  $(\rho_M - \rho_0)/\rho_0 = 0.019$ , which enhances  $U_g$  by  $\sqrt{6.3} \simeq 2.5$ . This value of the relative density difference is still small, well within the condition for the Boussinesq approximation, so it comes into play only through the reduced gravity g'. The baroclinic velocity is then reduced by a factor 20 with respect to the ocean. Since the horizontal scale is reduced by a factor 25,000, the time scale  $L/U_g$  is reduced by a factor 1,250.

To reproduce the Coriolis effect we need therefore to increase the Coriolis parameter f by a factor 1,250. Then the Rossby number  $Ro = U_g/(fL)$  is preserved from the ocean value. With a ocean value  $f = 0.8 \cdot 10^{-4} \, \mathrm{s}^{-1}$ , we set  $f = 0.1 \, \mathrm{s}^{-1}$  in the laboratory, corresponding to a tank rotation period  $T_c = 4\pi/f = 124.8 \, \mathrm{s}$ .



For the representation of the Mediterranean outflow in the Gulf of Cadiz, the relevant non-dimensional number is the Burger number  $Bu = (R_D/L)^2$ , where  $R_D = LRo/Fr = (g'H_s)^{1/2}/f$  is the Rossby deformation radius. Note that this definition corresponds to  $Bu = (Ro/Fr)^2$ , from which follows that Bu is automatically preserved since both Fr and Ro are preserved. The only problematic issue for the correct representation of the propagation of Mediterranean waters in the Gulf of Cadiz could be the turbulent friction (mixing) which may not be reproduced correctly in the relevant sites, such as at CS, due to the stretching factor of 10. This will be further discussed in subsection 2.3.

A further time scale is given by the tidal period  $T_{\rm tide}$  which must be also scaled by a factor 1,250. The semi-diurnal tide M2 with period 12hr25mn then results in a tidal period of 35.77 in the laboratory. The main effect of the tide is a time oscillating (barotropic) velocity in the Gibraltar Strait with amplitude  $U_b \simeq 1$  m/s, which gives a second velocity scale after the baroclinic velocity  $U_g$ . It must be scaled in the laboratory by the same factor 20 as the baroclinic velocity  $U_g$  (cf. table 1). The non-dimensional tidal amplitude  $A_{tide} = U_b/U_g$  is then preserved. This is adjusted by the oscillation amplitude of our plunger device. Note that the Strouhal number is the constructed using the barotropic velocity scale (cf. table 1). The propagation speed of the external tide  $(gH)^{1/2} \sim 50\,\mathrm{m/s}$  corresponds to a half wave length of the order  $1000\,\mathrm{km}$  so its phase can be considered as constant at the considered scale of the order of  $100\,\mathrm{km}$ . It is therefore well reproduced by our plunger device.

In summary, the laboratory model reproduces the key non-dimensional parameters except for the slope itself and the Reynolds number, see table 1.

#### 2.3 Viscous, turbulent and bottom friction

The Reynolds number is of course much reduced in the model, by a factor 50,000 (since the horizontal velocity is reduced by a factor 20 and the vertical scale by a factor 2,500). Therefore the reproduction of turbulent mixing and friction phenomena requires specific analysis. We must distinguish the interfacial friction between the two density layers and the friction in the bottom boundary layer.

Interfacial friction is related to entrainment at the interface. According to many studies Turner (1973) it becomes fully turbulent and independent of the Reynolds number beyond Re > 2,000, which is achieved in our experiments. In a self-similar steady regime, the entrainment velocity  $w_e$  depends on the slope angle by the empirical law of Turner (1973) (see also Ellison and Turner (1959); Beghin et al. (1981); Negretti et al. (2017); Martin et al. (2019))

$$w_e \simeq (0.005 + 0.06\alpha_s)u_H,$$
 (4)

where  $\alpha_s$  is the slope expressed in radians. As far as  $\alpha_s < 0.1$ , the turbulent interfacial friction does not depend on the slope, so that the slope rescaling  $\alpha_s \propto H/L$  does not modify the gravity current dynamics. The stretching factor of 10 imposed in the bathymetric model implies slopes larger than  $\alpha_s > 0.1$  at CS, so that a local smoothing in the region around CS has been applied to the topography (see section 3.1 for further details).

The bottom boundary layer is less prone to instabilities, and it remains laminar in many experiments of gravity currents on flat surfaces. In that case, the bottom friction force on a layer of thickness h can be estimated as  $u_*^2 \simeq \nu u_H/h$ , or  $u_*^2 = \sqrt{\nu f/2} u_H$  for an established laminar Ekman layer. The transition to a turbulent friction law occurs at  $u \sim 5cm/s$  in our experimental




conditions (Sous et al., 2013), so we are in a transitional regime, with possibly slightly higher friction. Moreover, our rough bottom is prone to local boundary layer detachment, leading to form drag effects. Over an obstacle of height  $h_d$ , the friction force is  $\propto c_D h_d u_H^2$ . On a series of obstacles separated by a distance  $\lambda_d$ , this yields an effective friction  $u_*^2 = c_f u_H^2$ , with  $c_f \sim c_D h_d / \lambda_d$ .

For the ocean, a turbulent friction law  $u_*^2 = c_f u_H^2$  is generally verified, with  $c_f \simeq 0.001$ . However, on very rough terrain leading to boundary layer detachment, we expect an effective friction coefficient  $c_f \sim c_D h_d / \lambda_d$ , according to the previous argument. For a steady current of thickness h submitted to such a bottom friction, we may write

$$h u_H du_H/dx = -c_f u_H^2 \tag{5}$$

The resulting relative velocity variation  $\delta u/u$  therefore scales like  $c_f L/h$ . Thus for a given friction coefficient  $c_f$ , the vertical stretching factor 10 leads to a friction effect reduced by the same factor in the experiment. This may be partly balanced by the enhanced viscous friction in the experiment, which increases  $c_f$ . For example, taking the laminar Ekman layer expression stated above, we get  $c_f = \sqrt{\nu f/2}/u$ . With our parameters, this gives  $c_f = 0.01$  for  $u = 2.2\,\mathrm{cm/s}$ , typically 10 times the ocean value. It then compensates the vertical stretching factor 10. In the case of a very rough terrain leading to boundary layer detachment, we argued that  $c_f \sim h_d/\lambda_d$ , which is increased by the stretching factor 10. A good similarity with the oceanic case is then expected.

Therefore we expect that turbulent friction effects can be reasonably reproduced in the laboratory, although not within the rigorous similarity of the inviscid equations in the hydrostatic approximation.

## 3 Experimental design

The experiments have been conducted in the Coriolis Rotating Platform at LEGI, Grenoble. The experimental setup incorporates realistic topography over an area equivalent to  $250\,\mathrm{km} \times 150\,\mathrm{km}$ , with a reduced vertical-to-horizontal stretching factor of 10, and includes barotropic and baroclinic forcing, as well as Earth's rotation. We used the traditional approximation neglecting the horizontal component of the Earth's rotation, as well as the variation of the Coriolis parameter with latitude, since, at the considered scales, effects of the planetary  $\beta$ -plane are irrelevant. Each forcing was calibrated so that it matched in-situ oceanic data reported in the literature, confirming dynamic similarity. A total number of 140 experiments have been realized, ( $\approx 40$  for calibration of each dynamical forcing and to check repeatability) using the same initial conditions but focusing on different regions.

#### 210 3.1 Realistic topography

The bathymetric model represents a region of  $150km \times 250km$ . For this, the SHOM bathymetry has been used (https://services.data.shom.fr/geonetwork/srv/fre/catalog.search#/metadata/LOTS\_BATHY), with an initial horizontal resolution of  $100\,\mathrm{m}$  smoothed to  $400\,\mathrm{m}$ , corresponding to a resolution in the laboratory of  $2\,\mathrm{mm}$ . The surface elevation of the topography has been corrected with the parabolic deformation of the free surface due to the rotation rate of the tank  $\Omega = f/2 =$ 

0.4782 rev/min, corresponding to  $h_b = h_0 + 0.5\Omega^2/g(r^2 - 0.5R^2)$ , with R = 6.5 m the radius of the Coriolis tank.

The origin of the axes is set at the CS summit  $(35.95^{\circ}\text{N}, 5.75^{\circ}\text{W})$ . To link the Cartesian coordinate system to Earth's latitude  $(\phi)$  and longitude  $(\lambda)$  system, the sinusoidal projection (also known as Sanson–Flamstedd) was used. The transformation relation is written as:

$$\lambda = \frac{180}{\pi} \frac{\sqrt{1 - e^2 \sin^2 \phi_{CS}}}{R_{eq} \cos \phi} x + \lambda_{CS} \tag{6a}$$





$$\phi = \frac{180}{\pi} \frac{\sqrt{1 - e^2 \sin^2 \phi_{CS}}}{R_{eq}} y + \phi_{CS} \tag{6b}$$

where  $R_{eq} = 6378.137 \,\mathrm{km}$  is the radius of the Earth at the equator, e = 1/149.38 a first-order eccentricity parameter of the Earth, and is  $(\phi_{CS}, \lambda_{CS})$  are respectively, the latitude and longitude of CS and are chosen as the origin of the right hand coordinate system. In the following, depending on the presented results, we will give both references to the experimental and real ocean scales.

Because of the vertical stretching factor of 10, slopes are enhanced, which may be problematic for non-hydrostatic effects, as well as regions of strong mixing like the CS area. Hence, we smoothed a region around CS of dimension  $(\Delta x, \Delta y) = (-0.4 \pm 0.1, -0.15 \pm 0.24)$  cm, corresponding to an area of 12.5km in East-West and  $10\,\mathrm{km}$  in North-South directions, respectively, in order to keep maximum slopes s below s 

Figure 1. Sketch of the experimental setup with the topography position inside the Coriolis tank, the position of the tidal generators and the gate position for the lock-exchange baroclinic initial condition. Dashed lines indicate the horizontal and vertical views of the optical measurements optical measurements and the positions of the lasers. The three green parallel lines in the strait represent the vertical sections for PIV/LIF measurements. Three capacitors measured the variation of the free surface at three positions in the Atlantic (SSH-A), in the Mediterranean (SSH-M) and at Tarifa (SSH-T). The dot off of Faro is a CTD used to detect the arrival of Mediterranean waters at the end of the topography. The bottom panels display an instantaneous picture of the topography installed within the Coriolis tank with a global view on the bathymetry of the Gulf of Cadiz (left) and an instantaneous image with horizontal laser prior to gate removal focused on the Strait viewed from the Spanish coast.




total depth of 65cm (9.6cm above CS) and put in anticlockwise rotation with a rotation period of  $T_c = 124.8 \,\mathrm{s}$ , as from the scaling given in table 1. One reservoir was filled with saline water with density  $\rho_M$ , representing the Mediterranean Sea, the second with a lighter fluid of density  $\rho_0$  corresponding to the Atlantic Ocean. The density difference was kept constant with  $\Delta \rho_0 = 0.19 (\pm 0.01) \, \mathrm{kg/m^3}$  in accord with the scaling given in table 1. Considering an emptying speed  $U_g$  and the half-section at CS, the experimental duration was estimated to be about 30 minutes, corresponding roughly to  $t/T_c \sim 15$  rotational days. At the start of the experiment following the gate removal, the flow through the strait undergoes through three regimes (Lawrence (1990); Zhu and Lawrence (2000); Negretti et al. (2007)): an initial unsteady phase in which the flow in each layer increases rapidly, a stationary regime called of maximum exchange and, finally, a sub-maximal exchange in which the flow decreases because no sufficient density difference is still present to maintain the initial hydrostatic pressure gradient. The useful phase for the experiment is the maximal exchange regime and this phase must be maintained long enough for the outflowing Mediterranean water to reach the end of the topography along the Portugal coast in geostrophic adjustment. Figure 2a shows the time evolution of the flow speed (blue continuous line) in the salty layer at CS at 1.6 cm (40 m in the ocean) from the bottom, where a stationary, maximal exchange flow regime is rapidly reached, with an average velocity of 6.9 cm/s. The speed resulting from applying  $G^2 = 2U_q^2/(g'H_s) = 1$  corresponds to  $6.8 \,\mathrm{cm/s}$ . The figure also shows that the baroclinic experiment lasts at least 25 minutes (21.7 days for the ocean) during the stationary maximal exchange. The red continuous line in figure 2a gives the salinity evolution from a CTD placed at the end of the topography  $(-7.830^{\circ}, 36.840^{\circ})$  1 cm (25 cm in the ocean) from the bottom along the Portugal continental shelf (see figure 1). It shows that the first saline water is detected after 7 minutes after gate removal in the Strait ( $\approx 6$  days in the ocean). This gives a stationary condition everywhere for 7 

Figure 2. (a) Time evolution of the baroclinic velocity (black line) measured using an ADV at the CS summit within the Mediterranean layer  $1.6 \,\mathrm{cm}$  ( $40 \,\mathrm{m}$  in the ocean) from the bottom, highlighting the rapid initial settling of the maximal exchange regime, which lasts up to 14 rotational days. The gray line is relative to a salinity measurement off of Faro (Portugal) along the continental shelf using a CTD-F probe  $1 \,\mathrm{cm}$  ( $25 \,\mathrm{m}$  in the ocean) from the bottom, highlighting the time for the Mediterranean Outflow needed to reach the Portugal coast at the end of the topography, corresponding to roughly 4 rotational days. (b) Purely barotropic velocities measured with the ADV at the CS summit within the Mediterranean layer  $2.6 \,\mathrm{cm}$  ( $65 \,\mathrm{m}$  in the ocean) from the bottom, being of maximum  $10 \,\mathrm{cm/s}$  in the spring tide configuration (black lines), and half in the neap tide configuration (green lines), with the corresponding SSH variations in the Atlantic (c) and in the Mediterranean (d) basins. (e) Comparison of the SSH anomaly over three tidal cycles for the Atlantic (purple line), the Mediterranean (orange line) and at Tarifa (cyan line), highlighting a phase shift between the Atlantic basin and the Tarifa station for the tidal amplitude of  $0.2T_{tide}$  in accord with observations (Roustan et al., 2024a). Experimental units (bottom-left), real ocean units (top-right).

tide configuration, which is in agreement with in situ measurements (Roustan et al. (2023)) considering a velocity scale factor of 20 (cf. table 1).

The barotropic speed is associated with a very low sea level (SSH) variation within the Strait and in the Mediterranean Sea. In the Gulf of Cadiz, on the other hand, the tidal amplitude can reach up to 1 m, but barotropic velocities are 10 times lower than






those observed in the Strait.

An interesting feature of the Strait of Gibraltar is that the barotropic flow, responding the standing-wave nature of the tidal sea level oscillation, heads west (tidal outflow) between low water and high water and heads east (tidal inflow) between high water and low water (García-Lafuente et al., 1990; Naranjo et al., 2015). This cannot be represented by our experimental set-up, in which an increase of sea level in the Atlantic corresponds to an eastward flow, and reversely during the SSH decrease.

The plungers can also reproduce the relative difference in SSH between the Atlantic and the Mediterranean basins, as their location in the western reservoir means that sea levels rise much less to the east of the strait than to the west. This is shown in figure 2(c,d), in which the SSH variations due to spring tide (red curves) and neap tide (blue curves) forcing are reported in the Atlantic (c) and in the Mediterranean (d). In the Atlantic basin, SSH variations are up to 2 mm for the spring tide forcing and half for the neap tide forcing, corresponding to 5 m and 2.5 m in the real ocean. This is not in similarity with the real ocean since we have not respected similarity of the external Froude number. The variations in the Mediterranean are much smaller, reduced by a factor 2.5 with respect to the Atlantic ocean SSH variations, in accord with observations (García-Lafuente et al., 1990; Candela et al., 1990). Since we are interested here in the dynamics related to the barotropic/baroclinic flow rather that the effects of the SSH, the consequences of these differences on the dynamics with respect to the real ocean case have no impact on the generality of our results.

Note that when combining the baroclinic and barotropic forcings, a refinement of the plunger's amplitudes has been performed, since the strength of the barotropic forcing leads to a different net baroclinic exchange flux across the Strait, as reported from observations. Indeed, the strong interaction of the tidal forcing with the rough topography leads to non-linear interactions between the transport and the density interface location, making the tide to contribute to the exchange flow at subinertial scale, via eddy-fluxes (Bryden et al., 1994; Tsimplis and Bryden, 2000; García-Lafuente et al., 2002a; Morozov et al., 2002). Figure 2e displays a zoom on three tidal cycles of the Atlantic (violet), Mediterranean (orange) and at Tarifa (cyan) SSH, highlighting a phase shift between the Atlantic basin and the Tarifa station for the tidal amplitude of  $0.2T_{tide}$  in accord with observations (Roustan et al., 2024a).

#### 3.3 Measurements techniques

Measurements consisted in both intrusive and optical techniques, which are detailed below.

Three Acoustic Doppler Velocimetry (ADV, Vectrino, operating at 240 Hz), four 125 MicroScale Conductivity and Temperature Instrument (MSCTI, PME Vista, California, USA) and seven digital 4-electrode conductivity sensors (Endress Hauser Memosens CLS82E) devices were used to monitor the velocities and densities in several regions along the Spanish/Portugal continental shelf and in several channels within the Strait, at the exit of the Strait and in the Gulf of Cadiz channels (e.g. Majuan Banks, Gibraltar Channels, Cadiz and Guadalquivir Channels and Gil Eanes Furrow). Since this paper focuses on the dynamics of the Strait in which we used optical measurements techniques, the detailed map and a table with the exact positions of the intrusive devices is given in Bardoel et al. (2026) only. The free surface elevation anomaly (SSH) in both the Atlantic and Mediterranean basins and close to Tarifa were monitored throughout the experimental runs using highly precise interferometers with a precision of  $10^{-2}$  mm.





Optical measurements were made using PIV in both horizontal and vertical views, combined with Planar Laser Induced Fluorescence (PLIF) for the vertical views in some experiments.

The PIV set-up consisted of a light source, light-sheet optics, seeding particles, several cameras, and PCs equipped with a frame grabber and image acquisition software. Polyamide particles (Orgasol) with a mean diameter of 60 μm and a specific density of 1.022 kg/m³ were added to both salt and fresh water compartments as tracer material for the PIV measurements. The laser provided a continuous light source, with the beam passing through an optical lens with an angle of 75° that diverged the laser sheet in the area of interest for the horizontal measurements. An oscillating mirror was used to produce the laser sheet in the vertical views experiments.

A set of experiments to capture the horizontal velocity fields in the Strait was run with the laser sheet coming from the Mediterranean side, using a high-resolution SCMOS camera (PCO) with a resolution of  $2560 \times 2160$  pixels. The laser system could be moved vertically along a linear axis to scan the water depth, yielding laser sheets for the Strait east of CS at five horizontal levels at z = -0.065, -0.10, -0.14, -0.17, -0.20 m scanned twice. For each plane, 1,400 images were taken at a frequency of 10Hz, corresponding to 140s measurement time for each plane.

Velocity fields were computed from PIV measurements using a cross-correlation PIV algorithm encoded with the UVMAT software (http://servforge.legi.grenoble-inp.fr/projects/soft-uvmat). Each element of the resulting vector field represents an area of roughly  $0.5\,\mathrm{cm} \times 0.5\,\mathrm{cm}$ . The maximum instantaneous velocity error is estimated to be  $\approx 3-5\%$ .

In the vertical configuration, the laser sheets coming from the top illuminated the full water column along the longitudinal section sketched in figure 3 in the Strait area, with an inclination of  $18.5^{\circ}$  North with respect to the East–West direction and passing through the CS summit. Images were recorded using a PCO camera at a frame rate of  $50\,\mathrm{Hz}$  to capture the velocity fields. The laser system could be moved horizontally along a linear axis to scan the cross Strait section, yielding laser sheets at three parallel longitudinal transects at a distance of  $5\,\mathrm{cm}$  and  $6\,\mathrm{cm}$  from South to North respectively, relative to the middle transect passing through the CS summit, as sketched in figure 3b.

For some of the experiments in the vertical configuration, the PIV measurements were simultaneous to PLIF measurements, for which an identical high-resolution SCMOS camera (PCO) with the same lens as for PIV was used, at the same frame rate of 50Hz, placed adjacently to the PIV camera and looking through the free surface with the same angle. An interferometer bandpass  $532\,\mathrm{nm}$  for PIV and a high pass filter with cut-off  $552\,\mathrm{nm}$  for PLIF were used to separate the emitted wavelengths for PIV and PLIF, respectively. The known initial concentration  $c_0$  of Rhodamine 6G dye was then diluted and thoroughly mixed in the Mediterranean compartment, whereas Atlantic water was mixed with Ethanol for refractive-index matching to deliver the 2D instantaneous velocity and density fields. The detailed calibration procedure for the PLIF measurements is given in appendix A. Rhodamine 6G was also used for flow visualization, especially to qualitatively track the pathways of the Mediterranean Outflow in the Gulf of Cadiz as used in Bardoel et al. (2026) and the interface in the Alboran Sea for internal waves, examined in Tassigny et al. (2026c).

**Figure 3.** On the left, an instantaneous image of the laser plane during the experiment comparing the real topography (red line), the smoothed topography used in the present experiments (blue line) and the topography after calibration as registered on the camera images (black shadow), for the central transect. On the right, sketch of the three transects considered for the PIV/LIF measurements (continuous lines), the middle one passing through the CS summit, and further southern and northern transects. The positions of the moorings MO2 and MO5 of the PROTEVS GIB2020 campaign (Bordois and Dumas, 2020) are given as well, along with the transects S2 and S3 (dashed and dotted lines, respectively), that will be used for comparison with the experimental data.

## 4 Dynamics in the Strait of Gibraltar



In this section, we present the results relative to the velocity and density fields, first as temporal average over seven tidal cycles, (section 4.1), then as an average sampled by the tidal phase (section 4.2). Since the flow is composed of a baroclinic (stationary) contribution, a periodic barotropic contribution and the fluctuating turbulent contribution, it is convenient to express the velocity and the density fields as the sum of the following three contributions (Hussain and Reynolds, 1970):

355 
$$[u,\rho](x,z,t) = [\overline{u},\overline{\rho}](x,z) + [\widetilde{u},\widetilde{\rho}](x,z,t) + [u',\rho'](x,z,t)$$
 (7)

The mean flow  $[\overline{u}, \overline{\rho}]$  is computed as a time average over 7 tidal cycles. The tidal oscillating contribution is computed with the phase average operator:

$$[\tilde{u},\tilde{\rho}](t) = \int ([u,\rho](s) - [\bar{u},\bar{\rho}])G(t-s)ds \tag{8}$$

where G is a Gaussian kernel with a standard deviation of  $0.2 ext{ s}$  (0.6% of  $T_{tide}$ ) and a period  $T_{tide}$ . The turbulent component 360  $[u', \rho']$  is obtained by subtracting these two averages to the raw signal.

In the following section 4.1, we first present time averaged velocity and density fields  $(\overline{u}, \overline{\rho})$ , whereas the inflow and outflow dynamics during both spring tide and neap tide  $(\tilde{u}, \tilde{\rho})$  are analyzed in section 4.2.

In the following sections, some experimental results are directly compared with in situ observations obtained during the PRO-TEVS GIB20 campaign (Bordois and Dumas, 2020) conducted by SHOM (Service Hydrographique et Océanographique de la Marine, the French Naval Hydrographic and Oceanographic Service). This large-scale intensive survey covered the Strait

of Gibraltar, the Bay of Cadiz, and the Alboran Sea during a strong (near-equinox) fortnightly tidal cycle in October 2020. In this study, we focus on a subset of measurements collected at moorings MO2 and MO5, which were equipped with Conductivity–Temperature–Depth (CTD) sensors used to reconstruct the vertical density field and an acoustic doppler current profiler. Data collected along transects S2 and S3 included velocity profiles measured with vessel-mounted Acoustic Doppler Current Profiler (VMADCP), conductivity and temperature profiles measured with moving vessel profiler as well as acoustic backscatter measurements. The instrumentation and their main characteristics are described in detail in Roustan et al. (2023). The locations of the moorings and transects are shown schematically in figure 3b.

### 4.1 Mean flow dynamics

385

395

## 4.1.1 Averaged velocity and density fields

Figure 4 presents characteristics of the mean velocity and density fields around CS averaged over seven tidal cycles during the stationary maximal exchange regime for the middle transect (cf. figure 3b) under three different tidal forcing scenarios: without tide (left column), neap tide (middle column), and spring tide (right column). The same for the northern and southern transects is presented in figures B1 and B2, respectively, given in the appendix B. The arrows in the top panels represent the in-plane mean velocities, whereas the colormap highlights the mean density anomaly distribution. Comparison of the top panels indicates that increasing tidal forcing leads to a thickening of the mixed layer on both sides of the sill. This effect is partly due to the time-averaging procedure, as the interface oscillates more strongly with increasing tidal amplitude, and partly due to enhanced mixing associated with stronger tides, as will be shown in Section 4.2, where tidal averages corresponding to maximum out- and inflow are presented.

The mean density fields shown in the top panels of figure 4 reveal that the time-averaged density is overall more diluted, by approximately 15–20% east of the CS, when tides are present, particularly during spring tide, compared with the purely baroclinic case (left panel). This will be also further discussed in section 4.2.

Also, it appears that Mediterranean waters are more diluted (about 30%) west of CS in neap tide compared to spring tide conditions, an aspect which will be further discussed in section 4.2. These observations are also applicable in the case of the northern (figure B1) and southern (figure B2) transects given in the appendix.

In the absence of tides, the layer of zero horizontal mean velocity  $h_u$  closely follows the pycnocline  $h_p$  on the eastern side of the sill, as illustrated by the grey line corresponding to  $\Delta \rho_0/2$ , presented in the middle rows panels of figure 4. The alignment between  $h_u$  and  $h_p$  is disrupted when tidal forcing is introduced, suggesting that tides generate entrainment of Mediterranean water to the west of CS. This aspect is further discussed in section 4.2 as well. Longitudinal velocities u averaged over the tidal cycles weaken when the tide is applied west of CS, compared to the purely baroclinic case (see second row panels of figure 4).

The abrupt downward plunge of the pycnocline and the increased thickness of the mixed layer west of CS suggests the presence of an internal hydraulic jump for all tidal conditions. This is supported by the strong vertical velocity  $\overline{w}$  change in the third row panels of figure 4 observed at the same location. When the tidal amplitude increases in spring tide conditions, the positive/negative maxima of the vertical velocity lower. This can be understood considering that the differences between inflow

Figure 4. Mean flow characteristics for the middle transect passing through the CS summit (cf. figure 3b), for the three tidal forcings: without tide, neap tide, and spring tide from left to right. The x axis represents the along-transect coordinate, being positive toward the east. The first row displays the mean density with the averaged velocities  $(\overline{u}, \overline{w})$  (white arrows). The second and third rows panels display the mean horizontal  $\overline{u}$  and vertical  $\overline{w}$  velocities, respectively, superposed with the  $\Delta \overline{\rho}/\Delta \rho_0 = 0.15$ , 0.5, and 0.85 (black, grey, and white lines respectively). In between, the composite Froude number  $G^2$  is displayed as a purple line. The last row panels display the turbulent kinetic energy  $TKE = (\overline{u'^2} + \overline{w'^2})/2$ . Experimental units (bottom-left), real ocean units (top-right).

and outflow are more pronounced during spring tide as compared to neap tide: during outflow, the Mediterranean waters are advected further downstream and the internal hydraulic jump oscillates west of CS with the tidal phase and disappears even during inflow, such that the averaged velocity fields are homogenized over the entire considered field on the west of CS. During neap tide instead, the internal hydraulic jump is stably localized close to the western flank of the sill and persists longer during the tidal cycle, even during part of the inflow. This will be further discussed in the following section 4.2 below.

The local composite Froude number  $G^2$  provides insights into the local criticality of the flow. Hereafter,  $G^2$  is defined assuming a two-layer flow, each layer of mean thickness  $\overline{h}_i$  characterized by a constant horizontal mean velocity  $\overline{u}_i$ , i.e.:

$$G^2 = \frac{\overline{u}_1^2}{q'\overline{h}_1} + \frac{\overline{u}_2^2}{q'\overline{h}_2}.$$
 (9)

TKE is preserved.

440

Mathematically, a condition of  $G^2 \geqslant 1$  is necessary for the development of a stationary shock such as hydraulic jumps (Armi and Farmer, 1985; Sánchez-Garrido et al., 2011). However, the condition  $G^2 \geqslant 1$  alone does not guarantee hydraulic control at a cross-strait section (Pratt, 2008). The analysis presented here remains local and restricted to each transect and focuses on the potential for the development of localized shocks. To compute the composite Froude number, a two-dimensional two-layer model has been defined, even if the validity of these assumptions is still largely debated in the community, as several studies emphasize that the mixed layer plays an important role in controlling exchange-flow dynamics (e.g. (Bray et al., 1995; Sannino et al., 2007)). We adopted the  $\Delta \rho_0/2$  isopycnal as the interface (Bray et al., 1995) and the average velocity and density (depth integrated) have been computed in each layer to determine the individual terms within the composite Froude number  $G^2$ .

The composite Froude number G², shown in figure 4 below the second row panels as a purple line, exhibits averaged values below unity everywhere and approaches unity at CS in all transects. Note that some uncertainty is present in the determination of G since velocity values close to the bottom boundary and the free surface are subject to more uncertainty and in some transects or tide conditions the upper layer velocities are sometimes missing close to the free surface. Hence, we expect our computed G² being rather underestimated. These effects combined with the three-dimensional nature of the flow, which calls for a global criticality of the flow to be defined, can explain the failure of the computed composite Froude number to capture regions in which G² ≥ 1 in the average. Note also that the presented values are a total temporal average: when computing the internal Froude number during a tidal cycle, values of G² ≥ 1 are captured, as it will be shown further below in section 4.2. The same conclusions are also valid for the northern and southern transects given in figures B1 and B2, respectively, that are given in the appendix.

The bottom panels of figure 4 (and B1 and B2 for the northern and southern transects given in the appendix) report the turbulent kinetic energy  $(TKE = (\overline{u'^2} + \overline{w'^2})/2)$ , which interestingly appears to be highest close to the bottom boundaries. The intensity of the TKE also increases with increasing tidal forcing and appears to be linked to the bottom boundary-layer detachment on the west side of CS and the formation of a recirculation area. In comparison, the TKE between the two layers at the interface of Mediterranean and Atlantic waters is roughly half as strong. Moreover, since the detachment appears only when the tidal forcing is present, as shown from the second row panels for the velocity u, the TKE is reduced close to the bottom boundary for purely baroclinic conditions and the highest values are found, instead, at the sheared interface. It should be noted that, due to the larger aspect ratio used in the experiments, the TKE is not strictly in similarity to that in the real ocean. Since the characteristic velocity scales are U in the horizontal and (H/L)U in the vertical, the relative contribution of vertical velocity fluctuations (w') to the TKE is enhanced in the experimental configuration, potentially leading to elevated turbulence levels. Nevertheless, because the vertical contribution remains smaller in magnitude than the horizontal one, the overall order of magnitude of the

An overview of the averaged horizontal flow within the Mediterranean layer in the Strait at different depths (from  $-142.5\,\mathrm{m}$  to  $-365\,\mathrm{m}$ ) is shown in figure 5, for the considered three tidal conditions (without tide, neap tide, and spring tide from left to right). Meridional and zonal velocities are displayed by the arrows, whereas the color fields indicate the vertical vorticity component  $\omega$ , normalized by the Coriolis parameter f. Overall, the flow is canalized toward the Tarifa narrow increasing the velocity while approaching CS, with similar velocity amplitudes in all depths except the deepest one at  $z=-365\,\mathrm{m}$ , for all

**Figure 5.** Averaged meridional and zonal velocities with superposed vorticity field (colormap) in the Strait of Gibraltar at various depths for the three tidal forcings: without tide (left column), neap tide (center column), and spring tide (right column).

three tidal conditions. Velocities are overall smaller in the purely baroclinic case. From the colored vorticity plots, it appears again that the highest values are reported along the coastline where strong changes of bathymetry (depth) appear, highlighted by the grey contour lines in figure 5, and in the region where the hydraulic jump appears immediately downstream of CS.

When approaching the bottom, the vertical vorticity is increasing overall within the displayed field especially for the spring tide conditions (left column) because of the increasing shear due to the the bottom topography and flow detachment.

Figure 6. Two-layer flow characteristics at CS for the planes(from left to right: northern, middle and southern transects) for the three tidal forcing. From top to bottom: pycnocline depth  $h_p$ ; zero horizontal velocity layer  $h_u$ ; vertically integrated horizontal velocity u for the Mediterranean (dashed lines) and Atlantic (dotted lines) layers; vertically integrated volume transport and below salt transport; two layer composite Froude number  $G^2$ ; bathymetry. Experimental units (bottom-left), real ocean units (top-right).

450

455

#### 4.1.2 Mean flow characteristics

The vertically averaged two-layer flow characteristics are summarized in figure 6 for the three tidal configurations and for the three transects: the left column corresponds to the southern transect, the central column to the middle transect, and the right column to the northern transect (cf. figure 3b). The bottom panels of figure 6 display the bathymetry relative to the three transects.

The first row of the figure presents the pycnocline depth  $h_p$ , defined as the position of the isopycnal corresponding to  $\Delta \rho_0/2$ , where  $\Delta \rho_0$  represents the initial density difference between Atlantic and Mediterranean waters. The pycnocline depth  $h_p$  shows minimal variation with changes in tidal forcing. In the northern plane, it remains essentially unchanged regardless of the tidal conditions. In the central plane, the tides cause a slight elevation of the pycnocline on the eastern side of CS, likely due to the thickening of the mixed layer, as predicted by hydraulic theory. This depth increase has been also reported in the numerical simulations of Sannino et al. (2007) and in observations Roustan et al. (2023). On the western side, the downward plunge of the pycnocline is less pronounced for spring tide conditions, likely as a result of the detachment of the Mediterranean waters from the bottom boundary during outflow. The zero horizontal velocity layer  $h_u$  exhibits limited sensitivity to tidal forcing as well, except near the sill where it becomes slightly shallower for spring tide, again likely linked to the boundary-layer detachment occurring during outflow. The third row panels of figure 6 shows the vertically averaged horizontal velocities in both the Atlantic (dashed lines) and in the Mediterranean (dotted lines) layers. Very little difference in these averaged values is reported for the different tidal conditions, except of lower values for the purely baroclinic case, as already reported in the literature (Roustan et al., 2023; Wesson and Gregg, 1994) and observed in the previous figure 5. In the Atlantic layer, velocities are increasing when moving toward the East due to the decreasing pycnocline depth and are nearby constant in all three sections for  $x > \approx 30 \,\mathrm{cm}$  ( $\approx 7 \,\mathrm{km}$  west of CS). In the Mediterranean layer velocities are increasing when moving to the West, reaching the highest values at CS, and then decreasing to a nearby constant value after the flow re-adjustemnt with the internal hydraulic jump for  $x > \approx 30 \, \mathrm{cm}$  ( $\approx 7 \, \mathrm{km}$  west of CS).

The local composite Froude number  $G^2$  computed from equation 9 and shown in the following row panels, remains subcritical throughout the sections as already mentioned above, but exhibits a noticeable increase approaching the sill location. The maximum value of  $G^2$  is approximately 50% higher during neap tide compared to the spring tide condition with exception of the southern transect where values remain similar for all tidal conditions. A local increase in  $G^2$  is also reported in correspondence of other topographical features at  $x = 30 \, \mathrm{cm}$  and  $x = 45 \, \mathrm{cm}$  West of CS, where other hydraulic controls have been suggested by previous authors (Wesson and Gregg, 1994; Izquierdo et al., 2001; Hilt et al., 2020; Roustan et al., 2023). The criticality of the flow varying with the tidal phase is further discussed in the following section.

Figure 7. Horizontal flow characteristics in the Strait of Gibraltar at  $z=-190\,\mathrm{m}$  for neap tide (left column) and spring tide (right column) conditions, at four tidal phases corresponding to outflow (first row), high water slack (HWS, second row), inflow (third row), and low water slack (LWS, fourth row). The vectors depict the zonal and meridional velocity components and the colour shows the normalized vorticity.





## 4.2 Dynamics in the tidal average

#### 4.2.1 Horizontal fields

Figure 7 displays the tidal average of the zonal and meridional velocities (arrows) together with the normalized vorticity field. We see that during inflow, the flow is arrested and slightly reversed within the Strait at this depth ( $z = -190\,\mathrm{m}$ ) and is clearly reversed during spring tide. The mean flow is directed following the continental shelf and influenced by the Coriolis force, resulting in canalizing the flow toward the North and the Spanish coast. We also see that very high values of vorticity (5 to 10 times the Coriolis frequency f) are concentrated along the coasts in proximity of the continental shelf characterized by strong depth gradients. Generally, during outflow (the top row panels) the northern coastline is characterized by negative vorticity values, whereas the southern coastline is characterized by positive vorticity, whereas during inflow (latter two row panels) positive vorticity dominates overall. Also, during inflow to outflow flow reversal (LWS), bands of positive and negative vorticity are present just east of CS, possibly suggesting the presence of internal waves.

### 4.2.2 Inflow, outflow and slack dynamics

Similar to the panels in figure 4 presented in the previous section, figure 8 shows the results from simultaneous velocity and density measurements, averaged over the phases of maximum inflow (first and second rows) and maximum outflow (third and fourth rows) for both neap tide and spring tide conditions. Here, we focus on the middle transect only.

The density fields displayed in the left panels show that during both maximum inflow and outflow, dilution is enhanced under neap tide conditions on both sides of the sill, consistent with the total averaged fields in figure 4, except during the spring tide maximum outflow, where the strong barotropic tide produces slightly greater dilution  $\approx 10\%$  east of the sill. A closer inspection, presented in figure 9, demonstrates that tidal forcing is capable of mixing the entire water column down to 500m east of CS during spring tide, with oscillations of the density as a function of the tidal phase of the order of 15-20%. A direct comparison between our measurements at the MO5 mooring and the in situ observations from the PROTEVS GIB20 field experiment confirms the presence of the same tide-phase-dependent oscillations and the full-depth density dilution east of the sill. A pronounced thickening of the pycnocline is also evident when tidal forcing is applied, compared with the purely baroclinic case in figure 4.

During maximum neap tide inflow, examination of both the density (left panels) and velocity fields (further right panels) shows that the Mediterranean flow can still surmount the sill, forming a thin layer that descends along the western flank of the slope. In contrast, under maximum spring tide inflow, the Mediterranean Outflow is blocked. The internal hydraulic jump disappears for both neap tide and spring tide maximum inflow conditions, as indicated by the purple line representing the composite Froude number  $G^2$  below the horizontal velocity field. Note that now  $G^2$  is computed taking into account the contribution from the mean and the tidal flow, i.e.:

$$G^{2} = \frac{(\overline{u}_{1} + \tilde{u}_{1})^{2}}{g'(\overline{h}_{1} + \tilde{h}_{1})} + \frac{(\overline{u}_{2} + \tilde{u}_{2})^{2}}{g'(\overline{h}_{2} + \tilde{h}_{2})}.$$
(10)

Figure 8. Mean flow characteristics for the middle transect passing through the CS summit (cf. figure 3b), for maximum inflow (first two row panels) and maximum outflow (last two row panels) for both neap tide and spring tide conditions. The x axis represents the along-transect coordinate, being positive toward the east. The first column displays the mean density with the averaged velocities  $(\overline{u}, \overline{w})$  (white arrows). The second and third columns panels display the mean horizontal u and vertical w velocities, respectively, superposed with the  $\Delta \rho/\Delta \rho_0=0.15$ , 0.5, and 0.85 (black, grey, and white lines respectively). The composite Froude number  $G^2$  is displayed as a purple line below the horizontal velocity u panels. The last column panels display the turbulent kinetic energy  $TKE=(\overline{u'}{}^2+\overline{w'}{}^2)/2$ . Experimental units (bottom-left), real ocean units (top-right).

Notably, larger  $G^2$  values east of the sill during maximum spring tide inflow suggest eastward propagation of the relaxed internal hydraulic jump, potentially contributing to the generation of internal solitary waves. This is even better seen in the




**Figure 9.** Hovmöller diagram at the position correspondent to MO5 mooring in the observational campaign GIB2020 displayin the relative density for the observational (top panel) and the experimental data of the HERCULES experiment (bottom panels). An oscillation of the order of 15-20% in the density in the full water column is evident in both panels.

slack dynamics presented further below, but is discussed in a separate paper focusing on internal solitary waves generation (Tassigny et al., 2026c).

TKE values are markedly reduced during maximum inflow for both neap tide and spring tide conditions compared with those in the total averages of figure 4. However, under neap tide conditions, since Mediterranean waters still pass over the sill, the strong shear in the thin Mediterranean vein close to the bottom boundary continue to generate localized moderate TKE.

During maximum outflow, very high TKE values are observed, concentrated in the bottom boundary layers west of the sill, with approximately half those values occurring along the sheared layer between the Mediterranean and Atlantic waters. The second-column panels show a clear offset between the pycnocline and the region of maximum velocity shear. Internal hydraulic control is evident at the sill for both neap tide and spring tide maximum outflow conditions, along with two additional control points farther downstream (to the west), more pronounced during spring tide. Vertical velocities are stronger and more localized under neap tide maximum outflow, indicating that the internal hydraulic jump extends farther horizontally under spring tide conditions.

A particularly noteworthy feature, previously visible in the total averaged horizontal velocity fields of figure 4 but now more clearly defined, is the region of weak or even negative velocities on the western flank of the sill, especially during spring tide outflow conditions. This pattern indicates that the Mediterranean vein detaches from the bottom boundary, which may be due to an adverse pressure gradient generated by the barotropic flow during outflow in the divergent region west of the CS. This divergent pressure field can become sufficiently strong to overcome the opposing baroclinic pressure gradient of the gravity current. This mechanism, which accounts for the elevated TKE values observed in the bottom boundary layers during outflow (see right column panels), is further examined in the following section 4.2.3.

Figure 10. Same as in figure 8 but for the slack dynamics: high water slack (HWS) and low water slack (LWS).

Figure 10 presents the dynamics at high-water slack (top panels) and low-water slack (bottom panels) for both neap tide and spring tide conditions. Overall, it can be seen that even at these tidal phases, neap tide conditions produce stronger dilution of Mediterranean waters west of the CS compared with spring tide, with a difference of roughly 20%. East of the sill, the highwater slack exhibits about 10% greater dilution overall than the low-water slack, with mean dilution values of approximately 80% and 90%, respectively.

During high-water slack, a thin layer of Mediterranean water continues to flow along the western flank of the sill (see horizontal velocity fields), while part of this flow forms a recirculation cell particularly evident under neap tide conditions in the first




panel of figure 10 and in the horizontal velocity fields of the second-column panels, where a circular region of positive velocity appears. Similar behavior has been observed in situ in Figures 9b and 12 of Roustan et al. (2023), who reported large overturns along the slope, suggesting a reversal of the Mediterranean outflow acting as a strong gravity current, in addition to the shear instabilities in the interfacial mixed layer previously described.

An internal hydraulic control, indicated by the composite Froude number ( $G^2 > 1$ ), is present at the sill for both neap tide and spring tide conditions during high-water slack, with an internal hydraulic jump farther west, as shown by the strong positive and negative vertical velocities in the third-column panels. The thin layer flowing along the western boundary of the sill is responsible for the high TKE values observed there (see first panels of the right column).

During high-water slack, a clear decorrelation between the pycnocline and the region of maximum velocity shear is visible in the second-column panels of figure 10 for both neap tide and spring tide conditions. The same holds true under spring tide conditions during low-water slack.

As already reported previously for the maximum inflow and outflow dynamics, tidal forcing induces a strong decorrelation between velocity shear and vertical stratification. Figure 11 shows vertical profiles of shear (S, continuous lines) and Brunt–Väisälä frequency (N, dashed lines) at the CS summit, corresponding to the position of the MO2 mooring, during four tidal phases: maximum outflow (first column), low-water slack (second column), maximum inflow (third column), and highwater slack (fourth column), under both neap tide (green lines) and spring tide (black lines) forcing conditions. The shear, S, is computed by accounting for all measured components of the velocity gradient, i.e.,

$$S = \sqrt{\left[\partial_x(\bar{u} + \tilde{u})\right]^2 + \left[\partial_z(\bar{u} + \tilde{u})\right]^2 + \left[\partial_x(\bar{w} + \tilde{w})\right]^2 + \left[\partial_z(\bar{w} + \tilde{w})\right]^2}.$$
(11)

As predicted by scaling arguments, the shear is primarily governed, to first order, by the vertical gradient of the horizontal velocity,  $\partial_z(\bar{u}+\tilde{u})$ . Since the tidally averaged density profiles are statically stable, the Brunt–Väisälä frequency, N, is computed directly from the absolute value of the vertical density gradient as:

$$N = \sqrt{-\frac{g}{\rho_0} |\partial_z(\bar{\rho} + \tilde{\rho})|}.$$
 (12)

Under neap tide conditions, during most tidal phases, the depth of maximum vertical density gradient is nearby co-located with the depth of maximum velocity shear. In spring tide conditions, instead, there is a permanent shift between the maximum velocity shear and the pycnocline, as highlighted by the difference between the black continuous and dashed lines in figure 11. During outflow, strong entrainment of Atlantic water shifts the shear profile upward, positioning the maximum velocity shear within the Atlantic layer. As the tide transitions from outflow to inflow, the direction of entrainment reverses, and the Mediterranean layer is advected eastward. Consequently, the region of maximum velocity shear is displaced downward, below the pycnocline, within the Mediterranean layer.

The frequent decorrelation between the region of maximum vertical density gradient N and the shear maximum S, implies that shear alone does not effectively mix the two water masses, since it is located within a layer of almost homogeneous water with a constant density. This explains why even with a stronger tidal amplitude, ST conditions are less effective in diluting water westward of CS compared to neap tide conditions. This observation also raises questions about the definition of

Figure 11. Vertical profiles of shear (full lines) and Brunt-Vaïsalla frequency (dashed lines) at MO2 during maximum outflow (first column), high waters slack (second column), maximum inflow (third column) and low water slack (fourth column) for neap tide (green lines) and spring tide (black lines) forcing.

the composite Froude number, since the layer depths  $h_1$  and  $h_2$  can be defined either from the density interface or from the velocity interface, leading to ambiguity. Defining the composite Froude number with a third layer could address this issue, but only if the intermediate layer is taken as the region between the maximum velocity shear and maximum density gradient, a choice that is uncommon, since the third layer is typically defined by intermediate density (Sánchez-Garrido et al., 2011). As during outflow, a clear detachment of the Mediterranean vein from the bottom boundary, driven by an adverse pressure gradient in the divergent region on the western flank of the sill, is evident during low-water slack for both neap tide and spring

28

tide conditions. This detachment again produces high TKE values in the bottom boundary layer. High values of TKE are also

585

590

present in the interfacial region between Atlantic and Mediterranean waters west of the sill (x 

Figure 12. Recirculation and flow detachment at CS. Top row panels display the barotropic speed  $u_b$  above CS for neap tide (left) and spring tide (right) conditions. The bottom panels display the correspondent contours of zero-horizontal velocity u colored as a function of the tidal phase, highlighting the presence of a recirculation area along the west bottom slope of CS during outflow.

contours of zero-horizontal velocity  $\overline{u} + \tilde{u}$  colored as a function of the tidal phase, highlighting the presence of a recirculation area along the western bottom slope of CS during outflow in neap tide and spring tide conditions. The tidal phase is expressed in terms of the vertical average of horizontal tidal velocity  $u_b$  above CS (top panels). Since the reversal of the tide is not syn-


chronized in both layers, this average gives slightly lower values than expected and previously measured. The detachment is characterized by contours of zero-horizontal velocities at a given phase of the tidal cycle and this is shown in the bottom panels of figure 12. During outflow (positive values of  $u_b$ ), detachment of the Mediterranean layer occurs, generating a reverse flow (eastward) on the western flank of CS. The detachment appears earlier in the tidal phase, persists over a longer time, and has a larger amplitude in spring tide compared to neap tide, consequently delivering stronger bottom turbulence.

According to Prandtl's boundary-layer theory, flow detachment occurs when an adverse pressure gradient is present. We consider a simplified two-dimensional model of the flow, consisting of two immiscible fluid layers separated by a pycnocline at a depth  $h_p$ . Detachment occurs when  $\partial_x p > 0$ , where x is the along slope coordinate. The total pressure gradient  $\partial_x p$  can be decomposed into three components:

$$\partial_x p = \partial_x p_b + \rho_0 g' \partial_x h_p + \partial_x p_{\text{nh}}, \tag{13}$$

where  $\partial_x p_b = \rho_0 g \partial_x \eta$  represents the hydrostatic barotropic pressure gradient driven by tidal forcing,  $\rho_0 g' \partial_x h_p$  is the hydrostatic baroclinic pressure gradient due to the tilt of the pycnocline  $h_p$ , which is always positive for a downslope gravity current, and  $\partial_x p_{\rm nh}$  denotes the non-hydrostatic pressure gradient. Assuming that viscous effects on the tidal flow are negligible, the barotropic pressure gradient can be approximated using Bernoulli's principle:

$$\partial_x p_b = -\rho_0 \partial_x \left( \frac{u_b^2}{2} \right). \tag{14}$$

This formulation clearly shows that deceleration of the tidal flow, such as that caused by a bathymetric divergence during outflow, leads to an adverse pressure gradient. Conversely, in regions where the tidal flow accelerates, the barotropic pressure gradient becomes negative, driving the motion of the flow. The sign of the non-hydrostatic pressure gradient cannot be determined from this simplified analysis, although its magnitude can be estimated. Considering L being the typical horizontal length scale, H the vertical length scale, H the vertical length scale, H the baroclinic speed scale, H the tidal speed scale and H0 the reduced gravity, the non-dimensional momentum equation in the bottom layer becomes:

$$\frac{du}{dt} = \left(\frac{U_b}{U_g}\right)^2 u_b \partial_x u_b - \frac{g'H}{U_q^2} \partial_x h_p + \left(\frac{H}{L}\right)^2 \partial_x p_{\rm nh}. \tag{15}$$

In our experimental setup, the barotropic pressure gradient is of order one, by design of the tidal forcing. Similarly, the baroclinic pressure gradient scales as the inverse of the squared internal Froude number, which is also of order one. Therefore, the barotropic and baroclinic pressure gradients are expected to be dynamically similar with the real ocean case, as the ratio  $U_b/U_g$  and the internal Froude number are preserved by the experimental design. The non-hydrostatic pressure gradient is two orders of magnitude smaller than the hydrostatic terms. Due to the tenfold increase in the aspect ratio in the laboratory compared to oceanic conditions, non-hydrostatic effects are expected to be more pronounced in the experiment. Nevertheless, they only influence the flow characteristics at second order and hence differences can be neglected.

Roustan et al. (2023) inferred strong turbulence near the summit of CS, and observed large overturns on its western flank further downstream during neap tide (compare their figures 8 and 11 in transects S2 and S3). These overturns extended down to depths of approximately  $\approx 70 \, \text{m}$ , significantly larger than those typically produced by shear instabilities, which were estimated








to be one order of magnitude smaller. They attributed these large overturns to the detachment of the Mediterranean vein from the western slope of CS during outflow conditions. Their explanation was based on the experiments of Baines (2008), who argued that on the western slope of CS, the gravity-driven flow behaves more like a turbulent plume than a smooth gravity current, resulting in substantial entrainment, rapidly mixing the current with the surrounding water, and causing it to intrude at an intermediate depth rather than to continue following the bottom topography. According to Wesson and Gregg (1994) and Roustan et al. (2024b) high turbulent dissipation rates during neap tide outflows were reported, comparable to those measured under spring tide conditions at the interface between the two layers. They attributed these elevated rates to the Mediterranean intrusion phenomenon proposed by Baines (2008).

Our results clearly indicate that the detachment of the Mediterranean vein is primarily driven by the adverse pressure gradient induced by the barotropic flow, rather than by intense mixing and subsequent intrusion of the Mediterranean vein, a condition present during both spring tide and neap tide phases. This barotropic component of the pressure gradient exceeds the gravitational forcing that drives the dense current downslope, ultimately causing the current to detach from the boundary. Under spring tide conditions, the strength of the barotropic flow leads to reattachment and deceleration of the Mediterranean vein further downstream promoting the transport (advection) of waters mixed through shear instabilities and the internal hydraulic jump further downstream as well. In contrast, during neap tide outflows, it seems that the shear turbulence at the interface, interacts with the boundary-layer turbulence induced by the detachment of the Mediterranean vein, as argued already by Wesson and Gregg (1994), as it can be observed from the TKE distribution and the combined velocity/density fields given in figure 4 (but also in 8 and 10). This effect, combined with a more co-localized positioning of maximum velocity shear and maximum vertical density gradient shown in figure 11, results in more localized (less advected) and intense dilution during neap tide conditions as compared to spring tide conditions.

These results therefore suggest the importance of boundary layers (bottom and coastal) in producing TKE and vorticity, which appears to be equivalent to or higher than that produced at the interface between the two layers because of shear. Boundary layers then contribute rather to energy dissipation than to mixing Atlantic and Mediterranean waters in particular during spring tide. These findings may explain why similarly high turbulent dissipation rates levels are observed under both neap tide and spring tide conditions in observational data (Wesson and Gregg, 1994; Roustan et al., 2024b). They also underscore the critical role that topography and bottom boundary layers (friction) play in governing the overall flow dynamics and turbulence and the strong interaction with shear-induced instabilities in promoting water mixing and dilution, an effect that, in some cases, can surpass that of interfacial mixing alone, as already reported in Negretti et al. (2017); Martin et al. (2019).

## 4.2.4 Hydraulic criticality

Figure 13 displays the Hovmöller diagrams of the pycnocline depth  $h_p$  along the three considered transects (south to north from the top to the bottom panels, see the bathymetry on the left) with superimposed the vertically averaged horizontal tidal velocity as a function of the tidal cycle, evolving from maximal outflow to maximal outflow through maximal inflow. The top panels give the variation of the SSH in both the Atlantic and the Mediterranean for neap tide (left) and spring tide (right) conditions. First of all, an increasing pycnocline depth in the Mediterranean at a given x position moving from the northern to


Figure 13. Evolution of the pycnocline depth  $h_p$  with the tide from the northern transect (top panels), to the southern transect (bottom panels) in neap tide (left column) and spring tide (right column) conditions. Black arrows indicate the vertically averaged horizontal tidal velocity. The bathymetry is given on the left. Experimental units (bottom-left), real ocean units (top-right).

the southern transect can be observed for both neap tide and spring tide conditions, due to the geostrophic slope induced by rotation within the Strait. A remarkable strong variation in time of the pycnocline depth is reported when passing from outflow to inflow ( $t \approx 8\,\mathrm{s}$ ) to the east of CS, suggesting that an internal bore is propagating to the East when the flow reverses. This is particularly evident during spring tide conditions in all three transects. Moreover, we also observe that the position of this front does not happen at the same time in the three transects: considering the positions of MO2 and MO4 (cf. figure 3b) a time shift of about 2s corresponding to roughly  $\approx 40\,\mathrm{mn}$  minutes in the ocean case is observed in the experiments. The trapped bore is released when the flow criticality is attained. Across-strait variations of the tidal current, stratification conditions, and topography can modulate the shape of the released bore. García-Lafuente et al. (1990, 2018) evidenced that the semi-diurnal tidal wave M2 propagates southward in the Strait of Gibraltar. The author exhibited a 15° lag for M2, which corresponds to 29 mn delay between North and South for the external tide. Sanchez-Roman et al. (2018) confirmed that a counter current


Figure 14. Evolution of the composite Froude number  $G^2$  with the tide for the northern (top panels), the middle (middle panels) and the southern (bottom panels) transects in neap tide (left columns) and spring tide (right column). White dashed lines delimit  $G^2 = 1$ . Experimental units (bottom-left), real ocean units (top-right).

heading east precedes the tidal signal in the central part of the Strait of Gibraltar. Thus, the tide tilts the internal bore south-eastward, a feature generally preserved in the far field as illustrated by a selection of SAR images presented in Roustan et al. (2024a). Note that differences in the values of phase shifts is attributed to the different considered locations in the different studies García-Lafuente et al. (1990, 2018); Sanchez-Roman et al. (2018); Roustan et al. (2024a) considering that some of them focus on the external tide instead of the internal tide as in our experimental study.

The fact that the results of the present physical model are in accord with respect to the reported phase shift between north and south, imply that this phenomenon is independent of the Kelvin wave propagating south to north in the real ocean case, and is unrelated to any barotropic effect, as in accord with the opposite tidal flow reversal with respect to the SSH elevation (Naranjo et al. (2014)).

During outflow ( $t < 10\,\mathrm{s}$  and  $t > 30\,\mathrm{s}$ ), we see strong variations in the pycnocline depth around CS, suggesting the presence of hydraulic control for both neap tide and spring tide conditions in all three transects. During inflow, these gradients of  $h_p$  are quite reduced, especially for the neap tide conditions, suggesting that the hydraulic control may have been lost at CS in all three transects. Note also the presence of quite high variations in the pycnocline depth in correspondence of the two other




topographical features at  $x = 30 \,\mathrm{cm}$  and  $x = 45 \,\mathrm{cm}$ , as mentioned in the previous section, for both the neap tide and spring tide conditions.

The composite Froude number of the tidally averaged flow (see equation (10)) is displayed in the Hovmöller diagram of figure 14 for both neap tide and spring tide conditions. SSH variations are given on the top panels, whereas the bathymetry is shown on the left panels. The values  $G^2=1$  are given by the white dashed lines. During neap tide conditions, hydraulic criticality is clearly present at CS for almost the entire tidal cycle, and is lost during the maximum amplitude of the inflow. This is consistent with the strong gradients of the pycnocline depth  $h_p$  and the strong variations in the vertical mean velocity  $\bar{w}$  reported in figures 13 and 4 (and B1 B2 given in the appendix B), even though the composite Froude number  $G^2$  of the mean flow as given in equation (9) remains below unity at CS when considering the overall time average. In the southern transect (top panel) an area of  $G^2>1$  to the east of CS during inflow appears which may indicate a possible propagation of a bore to the east. Also, as reported previously, we observe locally higher values of the composite Froude number ( $G^2=0.9$  in the middle transect at  $x=50\,\mathrm{cm}$ ) in correspondence of the other two topographic features to the west of CS. This is evident in spring tide conditions where hydraulic criticality is attained at CS and at both other two topographic features west of CS during outflow in all three transects. Also, a clear propagation of the bore to the east when control is lost at CS at flow reversal is evident. The time period over which the control is lost at CS during inflow is larger under spring tide conditions (from  $t\approx 5\,\mathrm{s}$  to  $t\approx 27\,\mathrm{s}$ ) as compared to neap tide conditions (from  $t\approx 10\,\mathrm{s}$  to  $t\approx 20\,\mathrm{s}$ ).

As noted above, the frequent misalignment between the pycnocline and the region of maximum shear across tidal phases (cf. Figures 4, 8, 10) calls into question the conventional definition of the composite Froude number. Layer depths,  $h_1$  and  $h_2$ , can be determined either from the density or the velocity interface, creating inherent ambiguity. Introducing a third layer could mitigate this issue, but only if it is defined as the region bounded by the maximum velocity shear and the maximum density gradient, a departure from the more common definition based on intermediate density (Sánchez-Garrido et al., 2011). Consequently, the composite Froude number must be applied with caution when evaluating hydraulic criticality or identifying control locations.

#### 4.2.5 Transports

Under the Boussinesq approximation, the mean mass transport per unit cross section can be separated into four components. The first corresponds to the advection of the reference density  $\rho_0$  by the mean flow, hereafter referred to as the volume transport. The second arises from the advection of the mean density anomaly,  $\Delta \bar{\rho} = \rho - \rho_0$  by the mean flow, and physically represents the transport of salt. The third and fourth correspond to the transports of the tidal and turbulent flow. Accordingly, the mean transport per unit cross section can be expressed as:

$$\int \overline{\rho} \, \overline{u} dz = \underbrace{\rho_0 \int \overline{u} dz}_{\text{volume transport}} + \underbrace{\int \Delta \overline{\rho} \, \overline{u} dz}_{\text{volume transport}} + \underbrace{\int \overline{\rho' u'} dz}_{\text{transport}} + \underbrace{\int \overline{\rho' u'} dz}_{\text{transport}}. \tag{16}$$

The volume, salt mean, tidal and turbulent transports are plotted from top to bottom, respectively, of the figure 15, for the three transects considered, north to south from left to right. The three tidal forcings are represented by different color's lines, as in the

**Figure 15.** Mean transport contributions for the three planes (from left to right: northern, middle and southern transects), for tidal forcing cases (Without tide, Neap tide an Spring tide). Volume transport (first line), salt transport (second line), tidal transport (third line) and turbulent transport (fourth line).

previous figure 6. The four terms are arranged from top to bottom according to the amplitude of their contributions to the mean transport, with the volume transport being the dominant component. Each subsequent term is at least one order of magnitude smaller.

In an idealized exchange flow without lateral variations, the volume transport (first row panels) should remain zero and constant along the strait axis. In contrast, the results show pronounced local variations at the main topographic constrictions, where strong cross-sectional velocity components develop to bypass the obstacles, as CS and those to the west. Overall, the transport is predominantly negative, indicating that, across the examined transects, a larger volume of Mediterranean water flows westward through CS than Atlantic water eastward. This imbalance is expected to be compensated by an enhanced transport of Atlantic water south of the transects, where the topography features an extended plateau. In addition, the geostrophic slope associated with rotation favors a greater westward transport of Mediterranean water in the northern part of the Strait (see figure 3b).

Figure 16. Evolution of the vertically averaged salt transport  $\int [\Delta \bar{\rho} + \tilde{\rho}] (\bar{u} + \tilde{u}) dz$  with the tide for the northern to southern transects from top to bottom panels in neap tide (left columns) and spring tide (right column) conditions. Experimental units (bottom-left), real ocean units (top-right).

The vertically averaged salt mean transport per unit cross section is displayed in the second row panel of figure 15. A significant decrease in transport is observed just upstream of CS for all tidal scenarios, suggesting again the presence of three-dimensional flow structures around the sill, with some water circumventing the sill rather than flowing past it. Mass transports are overall everywhere positive and decrease when moving from the southern to the northern transects because of the decreasing depth containing more diluted waters west of CS.

The tidal transport, shown in the next row of panels, exhibits distinct patterns under different tidal forcing conditions. During neap tide, the transport is negative east of the CS and approaches zero immediately west of the sill across all three transects. Under spring tide conditions, by contrast, the tidal transport remains close to zero both east and farther west of the sill, but




becomes increasingly positive from north to south in the vicinity of and just west of the sill.

The turbulent transport, presented in the bottom row of figure 15, remains near zero for both tidal regimes in the northern transect and for spring tide conditions in the middle and southern transects as well. However, it becomes negative approaching and west of the sill for the neap tide conditions. This negative sign indicates that more diluted waters are advected downstream during neap tide conditions compared with spring tide, for which the transport is dominated instead by the tidal rather than the turbulent component. Figure 16 shows the tidal-mean, vertically averaged salt transport over a full tidal cycle for neap tide (left column) and spring tide (right column) conditions. The volume transport component is not included, as we have shown in the previous figure 15 it is two orders of magnitude larger than the salt transport and primarily driven by transverse variability linked to the Strait's complex topography. Salt transport is again enhanced over the southern flanks, where water depth increases. The M2 tidal constituent modulates the mass transport, leading to a pronounced reduction of salt transport during neap tide inflow, and even reversing the direction to positive salt transport during spring tide inflow. Overall, tidal forcing promotes salt transport from the Mediterranean toward the Atlantic.

Previous studies (Candela et al., 1989; Helfrich, 1995; García-Lafuente et al., 2002b, a; Vargas, 2004; Vargas et al., 2006; Roustan et al., 2023) reported that the baroclinic flow varies inversely with tidal strength, corresponding to the fortnightly oscillation of the subinertial current induced by the modulation of tidal mixing. According to these authors, the modulation results from enhanced tidal mixing during spring tides, which increases turbulent viscosity and reduces vertical shear between the layers.

Using velocity data obtained from the experiment with transient tidal forcing (including the M2 and S2 tidal constituents), we observe a similar reduction in baroclinic flow to that reported by previous authors when considering the same location (MO2 mooring at the CS summit) and applying identical data filtering. This behavior is illustrated in figure 17.

The top panels display the horizontal velocity component, computed from the velocity data using an eighth-order Butterworth low-pass filter with a cutoff period of 95 s (35 hours), thereby retaining long-term and subinertial components. This processing follows the approach of Roustan et al. (2023), based on the methodology of Vargas et al. (2006). The velocity u is taken as the along-strait component of the velocity. It shows the temporal evolution of the filtered velocity throughout the analyzed interval and demonstrates excellent agreement between the transient tide experiment (left column) and with the PROTEVS GIB20 measurements (Bordois and Dumas, 2020) (right column). The  $\approx$ 30% reduction in net baroclinic horizontal flow with increasing tidal forcing from neap tide to spring tide is also consistent with earlier field measurements and modeling studies (Candela et al., 1989; Helfrich, 1995; García-Lafuente et al., 2002b, a; Vargas, 2004; Vargas et al., 2006; Roustan et al., 2023).

The second row panels presents the time series of the barotropic flow, marking three points in the tidal cycle corresponding to neap tide amplitude (magenta line), spring tide amplitude (red line) and intermediate amplitude (cyan line).

In the third line, we compare the tidally averaged steady-forcing experiments (without tide, solid lines; neap tide, dotted lines; spring tide, dashed lines) with the results of the transient-forcing experiment (colored lines) at three different time instants corresponding to the above mentioned varying tidal strengths. For the purely baroclinic case (solid black line), the lowest max-

Figure 17. (Top panels) Hovmöller diagram of baroclinic horizontal velocity for the transient tidal forcing experiment (including the M2 and S2 tidal constituents) (left column) and from the PROTEVS GIB20 campaign (Bordois and Dumas (2020)) (right column). The baroclinic velocities are computed from the velocity data using an eighth-order Butterworth low-pass filter with a cutoff period of 95 s (32 hours), thereby retaining long-term and subinertial components (Roustan et al., 2023; Vargas et al., 2006), in a vertical profile at the CS summit corresponding to the location of the MO2 mooring. Corresponding barotropic velocities  $u_b$ , computed as the vertical average of the horizontal velocity, are given below. (Bottom panels) Vertical profiles of horizontal baroclinic velocity for neap tide (magenta line), spring tide (red line) and intermediate tide (cyan line). Black lines give the mean horizontal velocity profiles for the steady-forcing experiments: without tide — solid lines; neap tide — dotted lines; spring tide — dashed lines.


imum velocities are found in the Mediterranean layer, consistent with the reduced vertical velocities. When a constant tidal forcing is applied, the maximum velocity decreases inversely with tidal strength. Similarly, in the transient-forcing experiment, the maximum velocity also decreases with increasing tidal amplitude in agreement with the in situ observations of the aforementioned authors (although the reduction is slightly weaker than in the steady-forcing cases in the experiments).

Horizontal maximum velocities decrease from approximately  $9.5\,\mathrm{cm/s}$  to about  $6.5\,\mathrm{cm/s}$ , corresponding to a  $\approx 30\%$  reduction from neap tide to spring tide tidal forcing. Moreover, the shape of the vertical velocity profiles changes systematically with the imposed tidal strength.

Three possible mechanisms could explain the reduction of net baroclinic flow with increasing tidal amplitude at the considered

**Figure 18.** Kinetic energy characteristics at CS for the three tidal conditions the three transect from north, to south (from left to right). First row panels display the vertically averaged mean kinetic energy (MKE, thick lines) and the vertically averaged tidal kinetic energy (PKE, dashed lines). The second row panels display the transfer rate between mean kinetic energy and tidal kinetic energy. Positives values indicate energy transfer from the tidal to the mean flow.

location. The first, proposed by earlier authors, attributes it to enhanced mixing during spring tide. However, we have shown earlier in this paper that mixing is, in fact, stronger during neap tide, making this explanation unlikely.

A second possible mechanism is the transfer of periodic (tidal) kinetic energy into the mean kinetic energy of the flow. These quantities, along with the turbulent kinetic energy (TKE), are shown in the top panels of figure 18 for the three analyzed transects (north, middle, and south, from left to right). The color scale indicates the increasing strength of the tidal flow, with darker shades representing stronger tides. In the northern and middle transects, the mean kinetic energy at the CS clearly decreases as tidal strength increases, whereas the opposite trend is observed in the southern transect. Moreover, outside the sill summit,

these tendencies exhibit strong spatial variability in both the transverse and longitudinal directions.

The bottom panels of figure 18 show the conversion rate between periodic kinetic energy and mean kinetic energy, given by  $\overline{u}\overline{w}(\partial_z\overline{u}+\partial_x\overline{w})+\overline{u}^2\partial_x\overline{u}+\overline{w}^2\partial_z\overline{w}$  in  $[W.kg^{-1}]$ . For tidal flow extracting energy from the mean flow (i.e., decelerating it), this conversion term should be negative. However, the figure shows it is positive everywhere, especially around the sill, indicating that the tidal flow transfers energy to the mean flow, thereby accelerating it. This trend intensifies with tidal forcing: the curve corresponding to spring tide conditions (black line) exceeds that for neap tide (purple line) near the sill. Consequently, the observed decrease in net baroclinic flux at the sill cannot be attributed to tidal-to-mean energy transfer, as our results demonstrate the opposite effect.

Instead, the analysis along and between transects reveals pronounced variability in both the longitudinal and transverse directions, underscoring the strongly three-dimensional nature of the flow. When the tidal forcing is strong, the flow tends to bypass the obstacle laterally rather than pass over it. This conclusion is consistent with the computed transports shown in figure 15, which display increasingly large transport values as the tidal forcing decreases at the CS summit, while also exhibiting pronounced spatial variability in both the longitudinal and transverse directions.

Therefore, we conclude from these considerations that fluxes and transports cannot be accurately derived from horizontally or vertically averaged fields. The significant spatial heterogeneity makes full three-dimensional budgets indispensable for correctly quantifying these exchanges and for drawing robust physical conclusions.

## 5 Conclusions




In this study, we present the design and implementation of a large-scale physical model representing the Strait of Gibraltar and its surrounding regions. The selection of this key area in ocean dynamics is justified by its intense, small-scale, non-hydrostatic processes, which have significant feedback effects on larger-scale circulation patterns in both the Mediterranean Sea and the North Atlantic Ocean.

The novelty of this work lies in the unprecedented level of realism achieved in a laboratory setting. The model incorporates the realistic topography of a  $250km \times 150km$  area, maintains a limited horizontal-to-vertical stretching ratio of 10, and includes both barotropic and baroclinic forcing as well as the Coriolis effect due to Earth's rotation. Utilizing advanced measurement techniques, we captured essential flow characteristics, including velocity and density fields, which confirmed the dynamic similarity of our model to oceanic observations Roustan et al. (2023); Gasser et al. (2017).

Through combined velocity and density measurements across vertical transects in the Strait and near the critical region of Camarinal Sill (CS), we reveal the dominant role of bottom boundary layers. These generate significantly higher levels of turbulent kinetic energy compared to the sheared interface between Mediterranean and Atlantic waters. We attribute the enhanced impact of bottom boundary layers to the detachment of the gravity current from the sloping topography west of CS, induced by an adverse pressure gradient during outflow, when superimposed with tidal flow. This provides a new explanation for the previously debated detachment of the Mediterranean vein observed via moorings along the western flank of CS Baines (2008); Roustan et al. (2024b), attributing it to boundary layer separation rather than due to the gravity current intrusion after enhanced






interfacial mixing produced by shear instabilities or the internal hydraulic jump as proposed by previous authors.

The detachment becomes even more pronounced during spring tides, leading to advection of the diluted waters further westward beyond CS. In contrast, during neap tides, the turbulent kinetic energy from bottom boundary layers coincides with the turbulent kinetic energy at the interface between the two layers. This dual mechanism may explain the high dissipation rates observed in-situ during neap tides Wesson and Gregg (1994); Roustan et al. (2024b), an issue which is investigated in detail in a forthcoming paper Tassigny et al. (2026b). Overall, we observe up to 20% reduction in density west of CS when tidal forcing is active, compared to purely baroclinic conditions—underscoring the crucial role of tidal dynamics in promoting full-depth mixing, as opposed to solely stratified shear-induced interfacial mixing.

We also evaluated the composite internal Froude number, using temporally averaged velocity and density data under the assumption of a two-dimensional, two-layer flow. Results from the overall time averaged analysis reveal that relying solely on the criteria  $G^2=1$  is insufficient to identify internal hydraulic criticality. Instead, critical conditions are better indicated when additionally abrupt changes in vertical velocity, sharp pycnocline plunges, and local thickening of the mixed layer coincides. When averaged over the tidal cycle, instead, values of  $G^2\geqslant 1$  are consistently found at CS during outflow for both neap tide and spring tide conditions across all three considered transects. However, during spring tide, hydraulic control is lost for longer periods during inflow phases and this loss of control propagates eastward. Additional regions of  $G^2\geqslant 1$  are identified west of CS, associated with strong pycnocline gradients and vertical velocity changes, suggesting multiple sites of hydraulic control out of CS during spring tide. For neap tide conditions, these features are present, but with  $0.5 




## Appendix A: Planar Laser Induced Fluorescence (PLIF) calibration procedure

In stratified flow experiments, accurately matching the refractive indices of the heavy and light fluids is essential for precise optical flow diagnostics. This is commonly achieved by using a saltwater solution as the denser fluid and an ethanol-water mixture as the lighter one. However, Daviero et al. (2001) showed that variations in refractive index can often be neglected if the salt and dye concentrations remain below specific limits: density differences under  $20 \,\mathrm{kgm^{-3}}$  and dye concentrations ( $c_0$ ) below  $70 \,\mu g/l$ . These conditions are satisfied in the present study, with  $c_0 = 5 \,\mu g/l$ , ensuring minimal optical distortion and reliable diagnostic results.

The Laser Induced Fluorescence (LIF) technique was employed to determine the density distribution across three vertical planes in the Strait of Gibraltar, coinciding with the Particle Image Velocimetry (PIV) measurement planes. Rhodamine 6G was introduced into the saline Mediterranean water at the onset of the experiment. This fluorescent dye exhibits an absorption maximum near  $530 \, \text{nm}$  and an emission maximum around  $550 \, \text{nm}$ . Upon illumination with a laser operating at  $532 \, \text{nm}$ , the dye emits fluorescence, the intensity of which is proportional to the local Rhodamine concentration. To account for the attenuation of laser light as it propagates through the fluid, corrections based on the Beer–Lambert law are applied. Accordingly, for a pixel representing a fluid volume located at position (x,z), with  $I_0$  denoting the incident laser intensity, c the Rhodamine concentration, which is indepedent of the pixel's position for the calibration procedure, and E the intensity of the emitted fluorescence recorded by the camera, the following relationship applies:

$$E(x,z) = c\alpha I_0 e^{-(a_w + \epsilon c)z} + \beta \tag{A1}$$

with  $a_w$  the absorption coefficient of water,  $\epsilon$  the absorption coefficient of Rhodamine and  $\alpha$ ,  $\beta$  adjustable constants (here we discard the absorption coefficient of salt and ethanol which can be neglected as suggested by Negretti et al. (2022)).

To determine the values of  $\beta$  and  $\alpha I_0$ , an in-situ calibration procedure was conducted. This involved capturing fluorescence images for six known Rhodamine concentrations. The calibration coefficients were then estimated through linear regression. It was assumed that the laser illumination at the free surface,  $I_0$ , is uniform across the measurement plane, and that each pixel of the camera sensor exhibits identical sensitivity to light. The latter assumption is justified by the fact that the image edges—where sensor sensitivity may vary—were cropped. Consequently, the parameters  $I_0$ ,  $\alpha$  and  $\beta$  are considered spatially invariant with respect to the (x,y) coordinates. As values for very low concentration are highly subject to noise, the linear regression is weighted by the coefficient of determination of the last linear regression (colorbar in figure A1a). We measured  $a_w = 3.2 \cdot 10^{-3} \, \text{cm}^{-1}$  and  $\epsilon = 3.7 \cdot 10^{-3} \, \text{cm}^{-1} \mu g/l$ , which values are very close to values measured by Negretti et al. (2022).

For each known Rhodamine concentration, a linear regression of the measured fluorescence intensity E on the camera sensor is performed independently at each pixel of the image. This procedure yields estimates of the calibration constant  $\beta$  and the combined proportionality term  $c\alpha I_0 e^{-(a_w+\epsilon c)z}$ , which varies with depth z. To extract the attenuation coefficient  $(a_w+\epsilon c)$ , a second linear regression is conducted on the logarithm of the proportionality term obtained from the initial fit.

The resulting attenuation coefficients, calculated across all horizontal rows of pixels and for each of the six Rhodamine concentrations, are shown in Figure A1(a). These values are subsequently fitted using a linear regression to determine the individual absorption coefficients  $a_w$  (for water) and  $\epsilon$  (for Rhodamine). Since measurements at very low dye concentrations




Figure A1. (a) Attenuation coefficient as a function of Rhodamine concentration. The colorbar indicates the determined coefficient  $r^2$  of each linear regression used to compute the attenuation coefficient. (b) Light E on the camera sensor against the corrected Rhodamine concentration  $ce^{-(a_w+\epsilon c)z}$ . Black lines represent linear interpolations.

are more susceptible to noise, the regression is weighted by the coefficient of determination from the preceding logarithmic fit, as indicated by the colorbar in figure A1(a). The calibration yielded absorption coefficients of  $a_w = 3.2 \cdot 10^{-3} \, \mathrm{cm}$  and  $\epsilon = 3.7 \cdot 10^{-4} \, \mathrm{cm}^{-1} \mu g/l$ , which are in close agreement with the values reported by Negretti et al. (2022). Figure A1b plots the light on the sensor E against the corrected concentration of Rhodamine  $ce^{-(a_w + \epsilon c)z}$  for each pixel of the image. A final linear regression is performed to compute the value of  $\alpha I_0$  and  $\beta$ . Knowing all the parameters, the concentration of Rhodamine can thus be recovered using the formula:

$$c(z) = \frac{[E(z) - \beta] \frac{e^{a_w z}}{\alpha}}{1 - \epsilon \int_0^z [E(z) - \beta] \frac{e^{a_w z}}{\alpha} dz}.$$
(A2)

Retrieving the concentration of calibration images from this formula, an error below 5% is found. Following a detailed analysis of the data, an additional correction was applied to account for the absorption of fluorescent light emitted by Rhodamine along its path to the camera. Figure A2 displays five representative density profiles: four located east of CS at  $x = -45.2\,\mathrm{cm}$ ,  $-35.2\,\mathrm{cm}$ ,  $-25.2\,\mathrm{cm}$  and  $-15.2\,\mathrm{cm}$  all corresponding to the outflow phase. In this figure, dashed lines represent uncorrected profiles, dotted lines correspond to profiles corrected for absorption along the vertical laser path, and solid lines show profiles corrected for both in-plane absorption and attenuation along the path to the camera. During the experiment, the imposed density contrast between Mediterranean and Atlantic waters was  $\Delta \rho = 19\,\mathrm{kgm}^{-3}$ . By applying both absorption corrections, the density profiles East of CS reveal a well-defined, uniform lower layer, with the density difference between the upper and lower layers closely matching the experimental conditions, as shown in Figure A2 by the vertical solid line. The correction of the absorption is also significant west of CS: although the absence of correction does not result in an apparently unstable profile, neglecting it would lead to an overestimation of the mixing intensity.


**Figure A2.** Effect of the LIF corrections on five vertical density profiles during outflow. Dashed lines represent no correction, dotted lines include correction for absorption along the laser paths, solid lines include additionally the absorption of light along the path to the camera. The vertical solid line indicates the initial imposed density difference between Mediterranean and Atlantic waters for reference.

## 900 Appendix B: Time averaged characteristics in the northern and southern transects at CS in the Strait.

Figures B1 and B2 show the velocity and density fields around the Camarinal Sill, averaged over seven tidal cycles during the stationary maximal exchange regime for the northern and southern transects, respectively, under three tidal forcing scenarios: weak tides (without tide, left column), neap tides (neap tide, middle column), and spring tides (spring tide, right column). In the top panels, arrows indicate the in-plane velocity vectors, while the colormap represents the density anomaly distribution. Horizontal (u) and vertical (w) velocity components are displayed in the second and third rows, together with iso-density contours shown as black and grey lines. The composite Froude number,  $G^2$ , is indicated by the purple line, and the bottom panels display the turbulent kinetic energy (TKE).

The main features observed, namely, density dilution enhanced in neap tide conditions, vertical displacement between maximum density and horizontal velocity gradients, boundary layer detachment of the Mediterranean flow on the western flank of 0 CS, and elevated turbulent kinetic energy near the bottom topography relative to the sheared interface, are consistent across the northern and southern transects and with the results presented in Section 4.1.

Figure B1. Mean flow characteristics for the northern transect passing north of the CS summit (cf. figure 3b), for the three tidal forcings: without tide, neap tide, and spring tide from left to right. The x axis represents the along-transect coordinate, being positive toward the east. The first row displays the mean density with the averaged velocities  $(\overline{u}, \overline{w})$  (white arrows). The second and third rows panels display the mean horizontal u and vertical w velocities, respectively, superposed with the  $\Delta \rho/\Delta \rho_0=0.15,\,0.5,\,$  and 0.85 (black, grey, and white lines respectively). In between, the composite Froude number  $G^2$  is displayed as a purple line. The last row panels display the turbulent kinetic energy  $TKE=(\overline{u'^2}+\overline{w'^2})/2$ . Experimental units (bottom-left), real ocean units (top-right).

Figure B2. Same as figure B1 but for the southern transect.

Data availability. Datasets are available upon request to the corresponding author.

Author contributions. M.E.N. designed the laboratory experiments. A.T., L.G. and J.S. participated in the design of the experiment. M.E.N.,
 A.T., S.V. and T.V. conducted the laboratory experiments. A.T., M.E.N. and S.B. processed and analyzed the experimental data. A.T. and
 M.E.N. performed the discussion and interpretation of the results and wrote the manuscript. S.B., L.G., J.S., L.B., X.C. contributed to the further discussion and interpretation of the results.

Competing interests. The authors declare no competing interests.

Acknowledgements. This work has been supported by the French National Research Agency in the framework of the "Astrid" program under contract number 22-ASTR-0005-01. We are grateful to J.-B. Roustan for fruitful discussions.

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
