# Peer review of "A realistic physical model of the Gibraltar Strait"

_EGUsphere, 2025_

## Referee Comment (RC1)

Review of "A realistic physical model of the Strait of Gibraltar" by A. Tassighy et al. A very interesting paper that presents the best laboratory physical model of the Strait of Gibraltar ever made so far. Congratulations to the authors.

The experiment provides such an amount of experimental data that authors face a serious problem to select which ones are going to be presented in the manuscript, which one is the logical sequence to show the results, how are they connected to each other and so on. The final manuscript focus on tidal dynamics for different tidal forcings and compare the results with those already published in papers that deal with the same issue, either from observational or numerical approaches. There are a large number of experimental data related to high-frequency dynamics that are not addressed here (they will be covered in other ongoing articles) for obvious reasons of length. The analysis of the tidal regime alone has resulted in this extensive manuscript, which will be read with great interest by the oceanographic community working in this very special environment.

The manuscript should be published in OS, although authors must revise it with care. Below is a list of (mostly) minor or very minor points, whose extensive length is a consequence of the interest with which I have read the manuscript.

Comments to the manuscript (following the order of writing)

L35. "*the Mediterranean Outflow Water strengthens the Atlantic Meridional Overturning Circulation*" I wouldn't dare be so categorical. There is still considerable debate about the role of the Mediterranean outflow in the AMOC.

L46. "*observationally-based*" rather than "*experimental*"?

L78-86. A short phrase stating the origin, orientation and positive direction of *x* and *y*-axes would be welcome. Origin is CS summit, as stated in line 216 later, but perhaps it is a good idea to mention it here. It seems rather obvious that orientation is along ("*x*") an across ("*y*") Strait, with positive directions eastward and northward, respectively. However, it is not until caption of Figure 4 (around L400) that this convention is explicitly said. The reason of this suggestion is that, here and there in the manuscript appear contradictions (writing errors?) regarding velocity and coordinate signs. Therefore, the convention must be clearly stated and this Section seems to be the right place.

Although not explicitly stated, the Atlantic and Mediterranean waters have the same temperature in all simulations, right? This makes density solely a function of salinity, which is important in certain parts of the work (i.e., transports). It would be appropriate to mention this at some point in Section 2.

L122. "*In the vertical momentum equation (3c)…*" equation (3c) is horizontal momentum isn't it?

L 135-136. "*and Hs $\simeq$ 100m is the half water depth at CS*" Actually, water depth at CS is nearly 300m. Why this scaling factor of 100m? Does it have to see with the mean thickness of the Mediterranean layer at CS?

L158. A "*s*" is missing. 35.77s

L163. The Strouhal number is new to me. Other readers may encounter the same difficulty of understanding what does it mean. A definition and relevance of this number (similar to the one given for the Burger number in lines 151-153) would be welcome

L174. " .. *on the slope angle*.." Bottom slope? Interface slope?

L223. Delete "is"

L229-230. References within brackets.

L242. 6% is a clear overestimate. Tidal transport exceeds 3Sv (if you refer to the amplitude of the oscillation). 1.5% - 2% is more realistic.

Figure 1. Nice Figure showing the core of the experimental team (and the infrastructure). I like it

L249. Are units of the density difference ("kg/m3") correct? In fact, 0.19kg/m3 is a very tiny difference, ten times smaller than in the actual ocean, which contradicts the statement of L143-144 about enhancing the density difference (and, also, units in the color scale of some figures, i.e., Fig.9).

L260-261. No continuous red line in Figure 2a. Grey line perhaps, as stated in the caption? And *y*-axis indicate density difference, not salinity. On the other hand "..*(25cm in the ocean)*.."  is 25m, right?

L276. " *… of the order of magnitude of 1m/s…*" Where? At CS? Please indicate.

L279. What is it meant by "*purely barotropic velocity*" here?. Does it refer to the amplitude of a purely barotropic tide, i.e., the tide resulting if both Atlantic and Mediterranean basins are filed with homogeneous water (no baroclinic forcing at all)?

Figure 2 caption. (L-3 says "*salinity measurement*" while *y*-axis indicate density difference. L7 mention SSH variations, L8 mention SSH anomaly. What is the difference between both SSHs? Apparently, they refer to the same process. L9-10 about the phase shift: see comment L305 below.

L291 No "red curves" in Figures 2c,d, but cyan or grey.

L305 A phase shift of $0.2T_{tide}$ between the Atlantic basin and Tarifa station is declared. Taking $T_{tide}$ the period of M2 (12.4h approx.) the shift would be 2.5h or 70º phase difference for M2 frequency. In the actual ocean, this phase difference is hardly 15º, much less than the one reported in the manuscript (consult harmonic constants in the interactive web https://portus.puertos.es of Puertos del Estado, SPAIN). In the real ocean the phase difference is caused by the Earth rotation in order to maintain geostrophic balance at tidal scale, achieved via changing the sign of the cross-strait free surface slope at tidal time-scale. Moreover, as mentioned in lines 285-289, the relationship between the sea-surface oscillation and the tidal barotropic current in the actual ocean cannot be reproduced in the laboratory. In the experiment, eastwards flow corresponds to rising tide in the Atlantic, just the opposite that in the ocean. That means that the Coriolis force causing the cross-strait slope (and, hence, the phase shift in Tarifa) acts in opposite direction in the real ocean and in the experiment, which invalidates the comparison of results. I suggest removing these comments both here and in the caption of Figure 2.

L379. When speaking of mean density anomaly, do you refer to the ratio $\Delta\rho/\Delta\rho_0$ that appears in the top panels of Figure 4  (where $\Delta\rho = \bar{\rho}-\rho_0$, $\bar{\rho}$ being the averaged density over the seven tidal cycles ($\Delta\rho_0$ is already defined), right?). If so, it is not a density anomaly, but a ratio of density differences (dimensionless). Assigning a symbol to this anomaly, which is frequently discussed in the text, would make it easier to read.

L388. When saying " ..*waters are more diluted (about 30%) etc..*", do you mean that the density anomaly  ratio is 0.3?

L390. "surface" instead of "layer"

L391. The grey line, according to Figure 4 caption, is $\Delta\rho/\Delta\rho_0$ = 0.5 (dimensionless), which corresponds to $\Delta\rho =\Delta\rho_0/2$ (units of kg/m3). Please modify so that text and figure caption do not disagree. Moreover, this specific value is used (implicitly) to define the pycnocline, as stated later in lines 413-414. In my opinion, what is said in these two last lines should be moved there (to the end of line 391, more or less).

L399. "*deepen*" or "*descend*" instead of "*lower*"?

Figure 4 caption. L3 "…*with the averaged velocities*…" Actually, it shows the vector sum of averaged *u* and *w* with the later largely exaggerated. The same applies to Figure 8 and also to Figures B1 and B2 in appendix B. L5 I guess you speak of the time-averaged composite Froude number, do you? Please indicate.

L405-406 The mean thickness computation in equation (9) requires the definition of an interface depth, which seems to be the pycnocline defined as $\Delta\rho/\Delta\rho_0$ = 0.5. This is said later in lines 413-414, but should be said before. Moving these lines (as I suggested above) will improve the text understanding.

L423-424. "… *given in figures B1 and B2 in the appendix*". Remove "*, respectively, that are given*"

L437. "*horizontal velocity*" rather than "*horizontal flow*". I have seen that the term "horizontal velocity" usually refers to the rotated (along-strait) component of the horizontal velocity throughout the text; particularly in the second half of the paper this seems to be the rule. But not always means this component. For instance, it does not in the aerial view maps of Figures 5 and 7, which actually show the "horizontal velocity". Perhaps it is convenient to state a criterion somewhere in section 2 or keep the notation "*u-component*" for the horizontal velocity in the second part (after presenting Figure 7). As you can see below in my review, there are a few comments addressing this point.

Figure 5 caption. L1 "*Averaged horizontal velocity*" instead of "*Averaged meridional and zonal velocities*". Insert "*relative*" before "*vorticity*"

Figure 6 caption. Are the presented plots time-averaged values? Please indicate. L3 "dashed" should be "dotted" and "dotted" should be "dashed". The sentence "*vertically integrated volume transport and below salt transport;*" has nothing to see with the panels. Please, remove.

L451-452. Remove the sentence "*The bottom panels …. three transects*" or, better yet, incorporate it at the caption of Figure 6.

L454. Remove "*where ……waters*". Unnecessary; it has been already said.

L466. "…*to the shallowing pycnocline..*" instead of "*…to the decreasing pycnocline depth…*"

L472-473. "… *50% higher during neap tide compared to the spring tide condition with exception of the southern transect where values remain similar for all tidal conditions*" This is not what can be deduced from Figure 6. Actually, it is in the southern transect where $G^2$ differences between spring and neap tides are the greatest. Maybe you are referring to the northern transect in the caption?

L474. "*x=-30cm and x=-45cm*" instead of "*x=30cm and x=45cm*"? (this has to see with my comment to lines 78-86 above). Nice to indicate the corresponding distances in km.

L480. "*horizontal velocity*" instead of "*zonal and meridional velocities*"

L480. "*tidal average*"? Such an average would provide a unique value for the averaged period, that is, for a tidal cycle, whereas Figure 7 presents snapshots at different moments of a tidal cycle. Maybe authors are referring to the average of the seven resolved tidal cycles for "*Outflow*", "*HWS*", etc.?

L486. "*…(latter two row panels)*" Do you mean "*third row*", which is where inflow is presented

L492. "*averaged over the phases of maximum inflow*" Again, do you refer to the maximum inflow averaged over the seven resolved cycles?

L497. "*demonstrates that tidal forcing is capable of mixing the entire water column down to 500m*". Not clear to me. I think you are referring to the periodic occupancy of the whole water column by Mediterranean waters (deep red colors in Figure 9). It could be mixing, but, in my opinion, is the uprising of the interface during the outflow which can eventually reach the surface during spring tides. If so, it is advection rather than mixing. A different issue is that, in HERCULES experiment in particular, the whole Mediterranean layer is well mixed from the bottom to the interface (which eventually can reach the free surface) with clear density differences between inflow (light red) and outflow (deep red) periods. These differences can be of the order of the mentioned percentage (15-20%, normalizing the density differences by $\Delta \rho_0$, I guess. Please, indicate)

L504. "*water can still overflow*" instead of "*flow can still surmount*"

Figure 8 caption. Please, indicate what the bottom panel is.

Figure 9. Please, explain what are the numbers given at the left and right sides of the color scale. I guess, numbers on the left are the $\Delta \rho$ difference in the actual ocean, and numbers on the right are the same for waters in HERCULES experiment. If so, note that the values of the order of 10 kg/m$^3$ that are representative of this scale are clearly contradicting (two order of magnitude greater) the values provided in line 249 for $\Delta \rho_0$. See my comment on L249 above as well.

L419. "*The second-column panels show a clear offset between the pycnocline and the region of maximum velocity shear*" The pycnocline ($\Delta \rho / \Delta \rho_0$ =0.5) is clearly seen in Figure 9. The region (surface?) of maximum shear is not so clearly depicted. It can be inferred by the color contrast (reddish *versus* bluish shading, probably), but locating it with a certain degree of accuracy is not an easy visual issue. West of Camarinal this color contrast (shear "interface") is located in a well differentiated depth range than the pycnocline. However, both surfaces coincide acceptably well elsewhere. Maybe the comment is referring to the west of Camarinal specifically? Please clarify the quoted sentence above.

L421-423. I have read the sentence "*Vertical velocities ..... spring tide conditions*" several times and I cannot catch its meaning and the reason to be here. Can you be a bit more explicit?

L525. "*even negative velocities*"? Do you men "*even positive*"? (cf, column #2, row#4 in Fig 8, better seen in column #2, row#4 of Figure 10)

L547-549. Same comments as in L419 above. I think the mismatch is limited to the west flank of CS.

L548. "*...under spring tide*". It rather seems "*neap tide*"

Figure 11 caption. Are vertical profiles just above CS? Please indicate

L572-573. "*...or from velocity interface*" Are you meaning "velocity interface (that is, $u$=0) or surface of maximum horizontal ($u$-component) velocity shear? The surface $u$=0 can eventually disappear during short periods of the tidal cycle and is useless for estimating layer thicknesses

Figure 12 caption. "*velocity*" instead of "*speed*"

L593. "*(positive values of $u_b$)*"? isn't it "*negative*"?

L656. Add "(*see the locations indicated in Figure 1*)" after "*conditions*"

Figure 13. As far as HERCULES experiment is unable to reproduce correctly the sea level phase relationship of the surface oscillation in the Atlantic and Mediterranean basins (lines 285-290 of the manuscript), it does not seem like a good idea to show the oscillation in both basins. I think the one in the Atlantic side is enough to delimitate the inflow and outflow periods in the Hovmöller diagrams below. I suggest to remove the Mediterranean sea level curve and add vertical dashed lines across the Hovmöller diagrams showing inflow and outflow periods.

Figure 13, caption. If possible, mark with dots the locations of M02 and M05 in the bathymetry left panel of the center section.

Figure 13, caption. "*vertically averaged horizontal velocity*" or "*vertically averaged u-component of horizontal velocity*"?

L660-661. "*Moreover, we also observe that the position of this front does not happen at the same time in the three transects*" I cannot see it in Figure 13. Perhaps some indications/marks in the very Figure would clarify the issue.

L662. I cannot identify the 2s shift mentioned in the text. Perhaps the marks I suggest above may help...

L670. "*García-Lafuente et al. (1990, 2018); Sanchez-Roman et al. (2018); Roustan et al. (2024a)*" within brackets.

L673. "*...with respect to the reported phase shift between north and south*" Are you speaking of the sea-level phase difference (i.e. barotropic tide)?

L680. "*x=-30cm and x=-45cm*"?

L691. "*x=-50cm*"?

L698. "*...or the velocity interface, creating...*" or " *...or the surface of maximum shear of the u-component of the horizontal velocity, creating..*"

L708. "*the transport of salt.*" This is true as far as *T*=cte. I guess *T*=cte has been implicitly assumed in the study. I have already made a comment on the interest of stating this constancy in Section 2. Nevertheless, it wouldn't hurt to mention it again here.

L709, eq.(16). Limits of integrals in the equation are *z*=0 and *z*=bottom, aren't they? To mention the limits in the text (or in the equation) would be OK to avoid any confusion with transports in either layer.

Figure 15 caption. "*Mean transport contributions…*" Add "*(per width unit)*" so that the sentence reads "*Mean transport (per width unit) contributions….*" On the other hand, does "*mean*" here refer to tidally averaged? Also write "*transects*" instead of "*planes*"

Figure 16 caption. Indicate what the black arrows are. (Transports per width unit, aren't they?)

L726. "..*positive*"? The mean (tidally-averaged?) transports in Figure 15 are mainly negative in all sections (north section could be a small exception). Figure 16 shows that the transport can be positive in spring tides in the eastern part, but only for short periods. But, when averaging over a tidal cycle, it appears to be *negative,* as seen in Figure 15. So what does "*positive*" refer to in this line?

**Lines 743-796. This part of the text should be a new sub-section 4.2.6 named "Time-variable tidal forcing amplitude" or "Transient tidal forcing" or something similar.** These lines clearly deal with a different topic than subsection "**4.2.5 Transports**"

L751. "*horizontal velocity component*" I think "*along-strait*" or "*u-component*" is meant (line 755). Please, correct. Same applies to Figure 17 caption.

L761. "*…three moments…*" instead of " *…three points…*"

L763. "*In the third row…*"

Figure 17 caption. Sentence "*with a cutoff period……. Vargas et al., 2006)*" is unnecessary as it is already in the text. I suggest a short "*(see text)*" instead. Add " *in the transient tide experiment*" after "*(cyan line)*" in line 7.

L766. "*… consistent with the reduced vertical velocities*"? No vertical velocities are presented. Is it meant "*…consistent with the reduced u-component velocity in the vertical profiles*"?

L771. A similar comment about the sentence "*…shape of the vertical velocity profiles*"

L773-796. This part of the manuscript must be revised carefully. There is mention to Figures/panels that are not seen in the manuscript (i.e. L778 mention a "*color scale*" in Figure 18, as if this Figure were a sort of Hovmöller diagram); line 773 indicates "*Three possible mechanisms*", when the text only addresses two; sentence in L786-789 "*Consequently, the observed decrease in net baroclinic flux* ==(flow?)== *at the sill cannot be attributed to tidal-to-mean energy transfer, as our results demonstrate the opposite effect*" Apparently contradicts the previous sentence in L785 "*…that the tidal flow transfers energy to the mean flow, thereby accelerating it*". The sentence "*the flow tends to bypass the obstacle laterally rather than pass over it*" in L790-791, when speaking, of a sill is confusing. How can a flow bypass laterally a sill? In my opinion, the flow either overflows the sill or remains stagnant in the upstream side

And an important question to be clarified, does the tidal flow in this final part have to see with the eddy-fluxes already discussed in several published articles in the literature. It seems that yes, but authors should clarify or comment the issue.

I have no posted comments to "*Conclusions*" section, as it should be partially rewritten by the authors in the light of the list of comments above. Neither have I revised Appendix A, as it addresses issues far from my expertise.

L904, Appendix B. "…*indicate the in-plane velocity vectors*" Actually it shows the vector sum of *u* and *w* components (with the later largely exaggerated) in the northern transect. The same applies to Figure B1 caption for the southern transect.

---

## Author Comment (AC2)

**Responses to Reviewer 2**

Review of "A realistic physical model of the Gibraltar Strait" by Tassigny et al.

*People love laboratory experiments. Even in an age where the smallest scales of turbulence are starting to become numerically resolvable, lab experiment give a sort of immediate physical connection that is hard to reproduce through simulation. It is good to know that the large turntable in Grenoble is being put to good use.*

*The particular experiment described in this paper is a scale model of the Strait of Gibraltar and immediately surroundings. The exchange flow is set up using a dam break scenario with topography that is realistic except for a scale factor of 10 and a bit of smoothing. Semidiurnal tides are imposed using oscillating plungers. Using modern laboratory measurements, including PIV, PLIF, ADV, conductivity and temperature sensors, and interferometers to measure surface elevation, the authors are able to gather data on velocity and stratification along three parallel transects of the strait. These are used to features at different phases of the tidal cycle and to compare conditions under spring and neap tide forcing.*

*For me the most interesting sections of the paper are those that describe detachment of the Mediterranean layer as it spills down the western flank of Camarinal Sill (CS) under conditions of outflow of the barotropic tide. The authors argue that strong barotropic outflow conditions create the adverse pressure gradient required for detachment and discuss consequences for mixing. Also interesting are insights into the role of bottom boundary layer processes in mixing, tidal rectification of the baroclinic flow and properties of the internal bore that propagates eastward. I suppose that some of the results on mixing between the upper layer and salty lower layer must be taken with a grain of salt since the Reynolds numbers are much lower than in the ocean.*

This important issue is dicussed in section 2.3 and in a forthcoming paper focusing on the high-frequency dynamics.

*Although the paper is lengthy, the narrative is generally easy to follow and there are just a individual points that need to be cleared up. I also have some suggestions regarding quantification of results that I hope the authors will consider. My recommendation is for moderate revision.*

We are really grateful to the referee for her/his thorough review and the positive comments. Her/his comments helped us refining certain discussions such as about the maximal/submaximal regime and on the composite Froude number.

A point by point answer to each of the referee's raised issue is given below in blue text. All changes have been highlighted by blue text in the revised manuscript.

**Main Points**

*One line 55 the text mentions "uncertainties" that remain and lists a few general categories such as internal hydraulics and instabilities, but it does not specify what the uncertainties are. The Introduction would be more helpful if it gave the reader more specific information about the issues that are unsettled and how a laboratory experiment (as opposed to a numerical simulation) will clarify or inform.*

We thank the referee for this remark. We have revised the Introduction to more clearly identify the unresolved issues and to explain why laboratory experiments are essential, stressing on our contribution. In particular, we now specify the 'uncertainties' we refer to, and emphasize the role of experiments in an era where numerical simulations have become powerful. Indeed, the representation of turbulent (non linear) processes, which control heat, mass and energy transfer, remain unresolved in numerical models and poorly constrained by observations. Inaccurate parameterizations can therefore lead to biased predictions and overlook the existence of multiple equilibria in climate systems. We clarify that large scale realistic laboratory experiments (such as on the Coriolis Platform) allow these processes to be isolated under controlled conditions, with systematic variation of key parameters and direct measurement of small-scale, non-hydrostatic dynamics, such as they appear in sea straits or in downwelling areas in the ocean. This approach provides physical insight and scaling laws that cannot be obtained from numerical simulations alone, where turbulence must still be parameterized, nor from field observations, where forcing and boundary conditions cannot be independently controlled. The resulting datasets provide a unique basis for improving our understanding of small-scale mixing, turbulence, and nonlinear processes that underpin spontaneous climate variability. These experimental

data are hence particularly valuable for improving numerical model predictions through more physically grounded representations of unresolved turbulent processes, training artificial intelligence (AI) approaches, and developing diagnostic tools for nonlinear processes in geophysical flows.

We have made the following changes at page 3

Despite significant advances in climate and ocean modelling, the accurate representation of turbulent processes governing vertical heat and mass transport remains a major challenge. Inadequate parameterizations can lead to biased simulations, missing key nonlinear feedbacks and multiple equilibria observed in geophysical flows (Danabasoglu et al. (2010); Rubino et al. (2020); Gacic et al. (2021); Pierini et al. (2022); Shi et al. (2022); Pirro et al. (2024)).

Observational [...] small-scale processes.

Large-scale laboratory experiments reproducing geophysical flows in dynamic similarity provide a complementary approach, allowing turbulent processes to be directly observed under controlled conditions, enabling independent variation of key parameters and measurement of small-scale, non-hydrostatic processes and their feedback on the mean flow by synoptic measurements, establishing robust scaling laws and physically grounded parameterizations.

Even if widely studied, several key processes remain under debate at the Strait of Gibraltar, including the persistence of hydraulic control at major topographic features (?Wesson and Gregg (1994); Bray et al. (1995); Pratt and Helfrich (2005); Pratt (2008); Sannino et al. (2009); Hilt et al. (2020); Roustan et al. (2023)), the observed detachment of the Mediterranean vein on the western flank of CS (Baines (2002); Roustan et al. (2024b)), the role of bottom boundary layers (Pratt (1986); Zhu and Lawrence (2000); Negretti et al. (2008, 2017)), the importance of 3D and non-hydrostatic effects (Zhu and Lawrence (2000); Pratt (2008); Sannino et al. (2009); Sánchez-Garrido et al. (2011); Sannino et al. (2014); Wirth (2025)), the quantification of turbulent dissipation rates and their unexpectedly high values measured during neap tides (Wesson and Gregg (1994); Roustan et al. (2024b)), and the generation mechanisms and properties of internal solitary waves observed in the Strait during neap tides (Roustan et al. (2024a))."

*The paper does not really discuss the topic of maximal vs. submaximal exchange. Maximal exchange stems from the Stommel and Farmer (1952 and 1953 FMR) papers on overmixing in estuaries. It has turned out that estuaries are not good candidates for overmixing, but as clarified in the Armi and Farmer papers (different Farmer), the Strait of Gibraltar appears to lie close to a state of maximal exchange, with two hydraulic controls, one at CS and the other in the Tarrifa Narrows (TN). This picture is complicated by tides and I think the current thinking is that the mean exchange is close to maximal and is perhaps pushed intermittently into a maximal state. The present experiments are quite interesting: Figure 14 suggests that at no point in the tidal cycle is the flow is hydraulically critical at both CW and in the Tarrifa Narrows (TN). However, during much of the tidal cycle the flow appears to be critical at one or the other. Generally speaking, maximal conditions mean that signals from outside of the strait are unable to propagate into and through the straight. (They are blocked but supercritical flow at each end.) When the flow is critical at CS, only signals from the west are blocked, and when the flow is critical at TN, only signals from the east are blocked. So this would appear to be inconsistent with maximal exchange, but I'll let the authors weigh in on what they think. Some comment should be made on this historical debate.*

We thank the reviewer for this valuable comment. Under lock-exchange initial conditions, following gate removal, the flow first undergoes a brief unsteady adjustment during which the velocity at CS increases with time. The system then enters a maximal exchange regime, so termed because the maximum possible transport through the strait is attained. Once the pressure gradient weakens and the interface in the Mediterranean layer drops below a critical depth, the exchange becomes sub-maximal. Following the seminal work of Farmer and Armi, numerous studies have addressed two-layer hydraulic exchange flows, including those by Greg Lawrence and others (?Zhu and Lawrence, 2000; Negretti et al., 2007; Prastowo et al., 2006; Fouli and Zhu, 2011).

During the experimental setup, we carefully monitored the velocity at CS over extended periods in order to capture its temporal evolution and to identify the stationary maximal exchange phase of the lock-exchange flow. This behavior is illustrated in the figure below for an experiment with higher density difference $\delta\rho = 25 kg m^{-3}$, where we clearly observe the loss of upstream control to the east at $t \sim 16$ min, followed by a gradual decrease in velocity thereafter. Note that the velocity during the steady maximal exchange phase is equal to $\sqrt{g' H_s/4}$ from the condition $G = 1$ at CS.

[Figure]

We have described this in the section 3.2.1 when describing the baroclinic forcing and have added now the following sentence at line 279 to further strengthen the message:

"During the maximal exchange phase, the strait is bounded by two hydraulic controls, which isolate the flow at CS from the time-varying boundary conditions in the two adjacent basins and thereby maintain a stationary exchange."

In general, as the referee correctly notes, critical conditions in two-layer exchange flows are reached either at constrictions, at sill crests, or at both. When only a sill or only a restriction is present, a so-called virtual control forms upstream with respect to the lower-layer flow, preventing upstream information from propagating into the region between the two controls, while the topographic control ensures blockage at the downstream end. Furthermore, there seem to be additional controls to the west, at Espartel sill and West Espartel site, the latter possibly permanent, which block any information propagating from the west toward CS. This is a separate question that can be addressed using our measurements conducted in this specific region. Consequently, no information can be transmitted between these two control points within the strait. When the exchange transitions to a sub-maximal regime, the upstream control is lost.

In our transects, we do not capture the TN location (by the way, precise localization of TN remains challenging, as its reported position varies by several kilometers in the existing literature), and it is true that the computation of $G$ does not suggest a clear control in the Tarifa Narrow in the purely baroclinic case when considering total averages, although it does when the tide is applied. Nevertheless, additional measurements (not presented here, as they were intended to investigate internal wave propagation) indicate that the interface in the Mediterranean basin descends with time. This outer-basin evolution does not influence the dynamics at CS, where control is maintained over most of the tidal cycle and where no external disturbances affect the flow. We do not know the exact location of the control point to the east, which also does not appear to be permanently present.

As noted by the referee, the presence of tidal forcing further complicates the system by allowing control points to vary spatially and temporally and to appear non-simultaneously, while CS remains dynamically isolated. Since perturbations propagate at the speed of internal waves, when a control emerges east of CS during inflow and the control at CS is lost, this information may not propagate fast enough, remaining trapped elsewhere without affecting the dynamics at CS. These considerations remain speculative at this stage. A detailed analysis is planned for future work using additional data covering the full transect from west of CS to the eastern exit of the Strait in the Alboran Sea, in both northern and southern transects. This will help elucidate the complex temporal and spatial system of hydraulic controls in the Strait.

We have also added the following paragraph at the end of section 4.2.4 as suggested by the referee:

"The question of the location and permanence of hydraulic controls in the Strait of Gibraltar is central to resolving the long-standing debate on whether the exchange flow is maximal or sub-maximal. If the Strait is bounded by two hydraulic controls, it is hydraulically isolated from both the Mediterranean Sea and the Atlantic Ocean. In that case, the exchange does not depend

on the pycnocline depth on either side of the Strait, but only on the local bathymetry and the density difference between the two water masses (Armi (1986); Armi and Farmer (1986); Lawrence (1993)). The Camarinal, Espartel, and West Espartel sills, as well as the Tarifa Narrows, are often cited as likely locations for such controls (Farmer and Armi (1988)).

These issues, however, are beyond the scope of the present paper and are therefore not discussed further here. Additional measurements of velocity and density, collected from west of CS to the eastern outer exit of the Strait, as well as over the Espartel and West Espartel sites and including both northern and southern channels, will allow a more comprehensive assessment of these questions and will be addressed in future work. "

*A third issue is the reliance of 1D metrics to characterize a 3D flow, a practice that continues in spite of the fact that theory has moved beyond the 1D setting. A case in point is the composite Froude number, which as the authors acknowledge, tells one something about local hydraulic criticality and says something about the ability of locally generated disturbances to propagate upstream. The local Froude number does is not by itself an indication of hydraulic control of the exchange flow as a whole. For example, if a hydraulic control exists at the CS section, it is because the entire baroclinic exchange across that section is being choked by the topographic constriction. If the width of the strait there were made to contract, or the topography to become shallower, a disturbance would be generated that would be felt across the whole width of the strait, would propagate into the Mediterranean, and would result in a diminished exchange rate and a deeper pycnocline. (A terrific educational video could be created if this exercise were set up experimentally and filmed.) In any cases, statements such as "hydraulic control has been lost at CS in all three transects" (line 678) don't make much sense: hydraulic control is not a local phenomenon or a property of a section. It would be quite easy to use values from the three sections to at least estimate the bulk criticality of the flow at any cross sectional using the generalized composite Froude number that the authors refer to in their citation of Pratt (2008, JFM). These take into consideration the velocity distribution and stratification across the whole section. If this value dropped below unity, the authors would be justified in claiming that control at CS has been lost. These remarks also apply in the vertical. The separation between the pycnocline and the level of maximum shear in some locations is nicely documented (e.g. Fig. 12). The authors remark that this separation clouds the use of a composite Froude number (2-layer or 3-layer). In situations like this, the local hydraulic criticality of the flow can be assessed by calculating the continuous vertical modes of the stratified shear flow using the Taylor-Goldstein equation or one of its extensions. This has been done in places like the Bab al Mandeb and Hood Canal (see Pratt, et al. JPO, 2000 and Gregg and Pratt, JPO, 2010) where it is sometimes difficult to identify a distinct density interface. I'm not suggesting that the authors do this now since the present manuscript is rather long, and the exercise of sorting through the modes can be a bit of work, but perhaps something to think about for the future. At one point, Bill Smyth had a nice code available through his Oregon State webpage. It has a provision for including viscous and diffusive effect in case there are critical layers. The results are nice because they give you wave speeds and provide a stability analysis. In general, I think modern investigators need to get away from composite Froude number is situations where the stratification and velocity are not "layered" and look at these modes.*

We thank the referee for this remark. We have already computed the composite Froude number along the three transects located at different cross-sections, as shown in figure 14 (and also in figures 4, 6, and those presented in the Appendix). These results show that, during inflow at CS, hydraulic control is lost contemporaneously at the three transects, although over different durations depending on both the transect location and the tidal strength. This led us to conclude that the control is lost across the entire cross-section at CS during these phases. In contrast, at other cross-sections located west of CS and associated with different topographic features in the Tangier basin, this simultaneous loss of control at all three transects is not observed during neap tides. For this reason, we do not further emphasize hydraulic control at those locations.

With regard to the referee's suggestion to compute the cross-strait variation of the composite Froude number following the methodology of Pratt 2008, we are concerned that this approach would require a number of strong assumptions regarding the cross-strait variations of both density and velocity. Given the limited spatial resolution of our dataset, namely, only three transects in the cross-strait direction and three transects in the horizontal planes, we do not believe that these assumptions would yield sufficiently reliable results.

Instead, we have performed additional computations of the composite Froude number using alternative definitions of the interface, in particular by considering the shear interface in addition to the isopycnal interface. We have also computed a three-layer composite Froude number, which is better suited to the structure of our data. The corresponding paragraph has been rewritten for improved clarity at lines 436, 445 and 724–746 (also in response to other referees' comments), and the additional Froude

number computations are now presented in Appendix D.

Furthermore, since the most reliable results, those that are consistent with the underlying assumptions used in the theoretical derivation of the composite Froude number, are obtained when using the local density difference rather than a constant initial density difference, we have replaced the original figures containing $G$ with those computed using the local density difference. Further details are provided in Appendix D.

We also acknowledge, as pointed out by the referee, that alternative approaches based on stability analyses (e.g. via the Taylor–Goldstein equation) would better account for the situations where the stratification and velocity are not "layered". However, we chose not to extend the discussion further in the present paper, as it would go beyond its scope. These aspects are instead addressed in a forthcoming paper currently under submission, which focuses on the generation and propagation of internal solitary waves within the Strait (Tassigny et al., 2026).

**Other Points**

*Figure 13 and discussion of processes east of CS: I'm curious about a certain aspect of the tidal flow in this part of the strait: namely the stripping of high potential vorticity water from the shallow shelf on the northern side, as seen in numerical simulations (see Dias, et al., JPO, 2025 Figs. 17 and 18). Water can be stripped away when the tide surges eastward and the advected plumes of high pv water lead to meanders in the Atlantic Jet. Do the authors see anything like this in the laboratory model?*

Indeed, our experiments reveal the presence of coherent vorticity structures detaching from the shallow coastal regions on both the Spanish and Moroccan sides (as seen from the vorticity plotted in the horizontal planes in figures 5 and 7, which are subsequently advected eastward during inflow conditions. To quantify these features more robustly, we rely on complementary high-resolution numerical simulations (LES) performed by our collaborators at SHOM and LOPS using the non-hydrostatic CROCO model, which faithfully reproduces the laboratory experiments conducted on the Coriolis platform.

The numerical simulations initially faced significant challenges related to turbulence closure, bottom friction parameterization, and the representation of complex bathymetry. These issues have recently been addressed through careful calibration and systematic comparison with our experimental data. High-resolution simulations are currently underway, and we expect that these results will allow a more detailed and quantitative characterization of the processes highlighted by the referee.

*Also, regarding the shallowing of the Atlantic layer along the north side of the Tarrifa Narrows (light yellow regions in Fig. 13). Armi and Farmer 1988, Fig. 11 show evidence that the pycnocline can intersect the surface, suggesting detachment of the Atlantic layer from the northern coast. Timmermans and Pratt (JPO, 2005) reproduce this using a rotating hydraulic model. Do the authors see this in the experiment?*

Even though the upper Atlantic layer can become very shallow during spring tides and strong inflow conditions (on the order of $\sim 12$ m), we have never observed the interface intersecting the free surface.

*I'm curious why are there no mention of Richardson numbers? Maybe mixing is discussed in the other paper.* Yes, estimates of the Richardson numbers are provided and discussed in a forthcoming paper addressing the high-frequency dynamics around CS (Tassigny et al.). We further show there that, due to the strong influence of boundary layers, the Richardson number is not a reliable indicator of mixing and dissipation in this context...

*The separation of the Mediterranean layer on the western flank of CS is nicely documented in Fig. 12. In many cases, the separation of a current from a rigid boundary is sensitive to details such as the boundary conditions imposed or the slope of the boundary. The fact that the slope in the experiment is magnified by a factor up to 10 may have some effect on the location of detachment (or lack thereof) of the descending overflow. Is this a concern?*

In Section 4.2.3, we explain the detachment of the Mediterranean flow along the western flank of CS and provide dimensional arguments showing that the different aspect ratio affects this dynamics only at second order. Of course, the exact detachment location may vary in the real ocean compared to our physical model, primarily due to the topography smoothing we applied in the experiment. Nevertheless, comparison with the in situ data of Roustan et al. (2023) confirms that the detachment occurs at similar depths and distances from the sill crest.

*line 32: outlet -> natural outlet. (don't forget the Suez canal)* Yes, the referee is right. We added 'natural' as suggested.

*lines 81-82. "The bottom topography is represented by the variable z=-hb(x,y)+h(x,y,t)...". This makes it sound as though the bottom topography is time dependent. Some clarification or perhaps a definition sketch needed here.*

230 The bottom topography is defined by the variable $h_b$, which does not depend on time but only on (x,y). But yes, saying that the bottom topography is represented by $z$ is wrong. It is represented by $h_b$, whereas the vertical coordinate $z$ is defined as specified. We have made the changes in the text.

*Lines 126-127 suggest that the term containing the external Froude number in (3b) can be neglected, but this term is clearly*
235 *much larger than the term containing the internal Froude number (Fr0 being small and Fr being O(1)). Perhaps the wording explanation is not clear?*
Indeed, stating that the term is negligible may be misleading, as it could be interpreted as being small in amplitude. In fact, the term is neglected because its dynamics occur on time scales that are much shorter than those of the internal processes of interest, which are characterized by larger temporal scales. Variations in SSH are therefore treated as instantaneous, implying
240 that the associated external dynamics are of higher order relative to the internal dynamics considered here. We have revised the sentence accordingly to avoid any misleading interpretation.
"However, since $Fr_0 \gg Fr$ and free-surface waves propagate much faster than the internal dynamics of interest, the term containing the external Froude number is of higher order and is therefore not further considered in the present analysis."

245 *Fig. 1 caption. "optical measurements" is repeated. Also, the photo in the lower left frame is of poor quality. There is lots of glare coming of the black bottom of the tank, making it difficult to see the actual topography. I wonder if the photo can be retaken with different lighting, or perhaps the photo can be edited to reduce the glare?*
We removed 'optical measurements' in the caption. The picture was included to provide a sense of the scale of the model. At this stage, it is not possible to take another photograph.
250

*Line 253. Delete "of".* Done.

*Line 271 "additional second" -> "second".* Done.

255 *Line 278. "ADV" has not been explained yet.* We added "(Acoustic Doppler Velocimetry, see section 3.3)".

*Line 285: "responding". -> "responding to"* Done.

*Lines 299-306. I had trouble understanding the thrust of this paragraph. It sounds like some sort of adjustment was made in*
260 *the plunger's amplitude to correct for some nonlinear process in the strait, but what is being corrected is unclear.*
We had to adjust the amplitude of the plunger (which is related to the displaced volume of water) to achieve the right mean baroclinic velocities in the strait, which are slightly modified if only barotropic forcing is applied without the baroclinic exchange flow. We have modified the paragraph to make this clearer.

265 *Line 315 I was unable to locate the Bardoel et al. (2026) reference and therefore unable to view the map.* We are sorry for this unprecision, the paper is in preparation as well, now it is specified in the bibliography.

*Line 390. "layer of zero horizontal mean velocity hu". Is hu the thickness of the layer, the depth of the layer, or just a label for the layer?*
270 $h_u$ is the depth of the layer at which the vertical shear of along strait velocity is maximum. We write clearly now in the text that we refer to the 'maximum vertical gradient of along-strait velocity'.

*Line 394. "when the tide is applied west of CS" -> If I understand the meaning it would be better to write "west of CS when the tide is applied".*Done.

275

*Line 398. "lower" -> "decrease"*Done.

*Lines 420-421. I'm not surprised or worried that G2 does not quite get to unity in the regions where the flow visually appears to be supercritical. I have encountered the same issue in other straits where there is clearly a locally subcritcal-to-supercritical*

280 *transition.*
We have modified now the definition of the composite Froude number using a local $g'$ as discussed above. Now, criticality is achieved in the mean in all three transects at CS for all tidal conditions.

*Line 430 states that the highest TKE values are found at the sheared interface for pure baroclinic conditions, but when*

285 *I look at the bottom left panel of Fig. 4 I don't see any elevated TKE at the interface. It looks instead that TKE decreases monotonically from the surface to the bottom. Am I looking at the wrong thing? Perhaps there is a tiny elevation of interfacial TKE to the east of CS.*
The high TKE values at the interface are observed at the onset of the descent. Further downstream, the interface becomes indistinguishable due to the development of the hydraulic jump, where TKE values increase substantially. During spring tides,

290 the hydraulic jump is displaced further downstream, allowing the interface to remain identifiable over longer distances toward the west. We have now clarified this point in the text.

*Fig. 6 caption. There is a reference to "vertically integrated volume transport and below salt flux" but I don't see either of those in the figure panels.*It was an error referring to an old version of the figure. It has been removed.

295

*Line 468. It is curious that G2 is maximum at CS and not immediately to the west of CS, where the Med layer spills down. Could this be due to the definition of the Med layer, which might include low-velocity contributions from the mixed wedge of water that lies between it and the Atlantic layer. One is probably justified in using a 1.5-layer Froude number for the overflowing layer in this case.*

300 The maximum also extends to the west of the sill crest, reaching approximately 2.5 km beyond the crest. Its extent is sensitive to the choice of interface definition, as illustrated in Appendix E of the revised manuscript, where we compare different methods for computing the interface for the determination of the composite Froude number.

*Line 495 claims that dilution is enhanced under neap tide conditions on both sides of the sill. Does this mean enhanced*

305 *compared to spring tide conditions? (When I look at the 3rd from top, left panel in Fig. 8, I don't see much dilution in the water column east of CS, and the dilution to the west of CS does not seem to be any greater than in the lower left panel.)* Yes, we mean enhanced compared to spring tide conditions. We agree with the referee that it is not easy to see this directly from the colorbar of the figure, but it is the case. Here we give vertical sections an both east and west sides of CS of the density anomaly for neap and spring tides, highlighting what we claim. We added this figure in the appendix C, and we added the following

310 sentence at line 415:
"Direct comparison of the mean dimensionless ratio of density differences $\rho^*$ at different position in the along-strait direction for the three tidal forcings is given in the appendix C".

*Fig. 9. The caption is a little unclear here. The bottom panel shows results from the experiment, correct? Then what is*

315 *HERCULES?* Yes, that is. We have rephrased the caption. HERCULES is the acronym of the experimental campain.

*Line 510. Why would "slightly elevated G2 to the east of the sill" suggest eastward propagation?*
The previous formulation was incorrect. Our intention was to provide additional clarification regarding the nature of the supercritical regions. Specifically, an eastward supercritical region exists, in which internal waves can propagate only eastward,

320 and a westward supercritical region, in which internal waves can propagate only westward. The hydraulic control located above the Camarinal sill therefore corresponds to a westward control, whereas the control forming to the east during inflow

corresponds to an eastward control. The distinction between these two types of control cannot be inferred from the composite Froude number alone, since $G^2 - 1 \propto c_+ c_-$ (Zhu and Lawrence (2000)), but instead requires knowledge of the sign of the characteristics $c_\pm$. Computing the characteristics explicitly is beyond the scope of the present paper; however, their sign can be inferred from the direction of the barotropic velocity. Consequently, the loss of control at the Camarinal sill during inflow relaxes the internal hydraulic jump and allows it to propagate eastward. We have therefore added the following sentence to clarify this point:

"The hydraulic control that forms above CS during the outflow phase is distinct from the control that develops during inflow to the east of the sill, in the direction permitted for internal wave propagation. The control at CS prevents the eastward propagation of internal waves, whereas the control located to the east blocks westward propagation. When the control at CS is subsequently relaxed, the internal hydraulic jump propagates eastward, potentially contributing to the generation of internal solitary waves. This behavior is more clearly illustrated in the slack-water dynamics presented below and in the Hovm"oller diagram of $G^2$ shown in Fig. 14. A detailed discussion of this process is provided in a separate study focusing on internal solitary wave generation (Tassigny et al., 2026)."

*Line 525. By "negative" (that is, westward) do you actually mean "positive" (eastward)?* Yes, we have made the signs corrections throughout the text.

*Line 534. Has "dilution value" been formally defined?* Following the suggestion of Referee 1, we have now defined the dimensionless density anomaly ratio $\rho^* \equiv \Delta\rho/\Delta\rho_0$.

*Lines 537, 545 and 637. When you write "along the western flank" many readers will think you mean parallel to the flank (or along-isobath), just as "along coast" means parallel to the coast. I think you are describing the flow that is spilling down the western flank of the sill, and you might consider phrasing it that way.* Done.

*Lines 521 and 544 speak of a control and hydraulic jump to the west. It looks like these occur near x=-45. Is this the Spartel sill, and are these features due primarily to the presence of this sill? Is there some reason it (and Tariffa narrows) is not referred to by name?*

As shown in Fig. 3b, the western end of the measured transects is located at the entrance of the Majuan Bank, but not yet at Spartel Sill. The eastern end of the transects lies close to, but does not reach, the Tarifa Narrows, which are located further east and extend so far we understood from the literature over several kilometers, e.g. there is not an exact location agreed from all the community as for Camarinal. Since no measurements are available there, this feature was not included in the original sketch. The Majuan Banc can be identified by the elevated plateau visible along the northern transect, although it is only partially captured at the western edge of the domain. The two topographic features referred to further downstream to the west of CS are not related to Spartel, they are present in the Tangier basin; however, we are not certain of their exact nomenclature.

In addition, combined velocity and density measurements were carried out in both the northern and southern channels around
355 the Majuan Banc, the latter including Spartel, downstream toward West Espartel. This specific dataset will be presented and
discussed in a forthcoming paper.

*Line 598. "x" is the zonal coordinate, as in Fig. 12. When you say that x is the along-slope coordinate, this seems to be*
*different. What does "along slope" mean? Without this understanding it is difficult to interpret eqs. (13) and (15). Are you*
360 *interested in the zonal component of pressure gradient, or the component of pressure gradient tangent to the topography, or*
*the x-component of pressure gradient tangent to topography? Precise mathematical definitions are nice.*
In the presentation of the results from vertical sections, the horizontal velocity always refers to along-strait horizontal velocity
as we are only measuring in-plane velocities. To avoid confusion with the horizontal planes presenting horizontal velocities,
we replaced the 'horizontal' velocity spelling of the measurements in the vertical sections with 'along-strait' velocity.

365

*I can see the internal bore in Fig. 13, and I think I can see the time shift, with the change in the pycnocline depth first*
*apparent at the southern transect and later at the northern transect.There is a later assertion that this time lag is independent*
*of Kelvin wave dynamics, but if the eastward propagating bore has some characteristic of a Kelvin wave, the signal along the*
*southern coast would tend to lead the signal along the northern coast. This was shown by Federov and Melville (JFM,1996),*
370 *who developed a solution for a "Kelvin bore" whose leading edge curves away from the coast and would, in the case of*
*Gibraltar, lead to the southeastward tilt cited on Line 667. Why is this not a possible explanation?*
Following the comments of Referee 1, who was not convinced by the presence of a time shift, we attempted to quantify this
effect more rigorously. In doing so, we found that the shift is not clearly identifiable for every tidal cycle and, moreover, that it is
not consistently equal to 2 s. Since this result is therefore not robust and cannot be demonstrated convincingly in a quantitative
375 manner, we decided to remove this discussion from the manuscript.

Well, I hope these remarks improve an already-nice paper. We thank the referee for these very pertinent remarks, which
significantly helped us to improve the clarity and overall quality of the discussion in the manuscript.

**References**

Armi, L.: The hydraulics of two flowing layers with different densities, Journal of Fluid Mechanics, 163, 27–58, https://doi.org/10.1017/S0022112086002197, 1986.

Armi, L. and Farmer, D.: Maximal two-layer exchange over a sill and through the combination of a sill and contraction with barotropic flow, Journal of Fluid Mechanics, 164, 53–76, 1986.

Baines, P.: Two-dimensional plumes in stratified environments, J. Fluid Mech., 471, 315–337, 2002.

Bray, N. A., Ochoa, J., and Kinder, T.: The role of the interface in exchange through the Strait of Gibraltar, Journal of Geophysical Research: Oceans, 100, 10 755–10 776, 1995.

Danabasoglu, G., Large, W., and Briegleb, B.: Climate impacts of parametrized Nordic Sea overflows, J. Geophys. Res., 115, C11 005, 2010.

Farmer, D. M. and Armi, L.: The flow of Atlantic water through the Strait of Gibraltar, Progress in Oceanography, 21, 1–103, https://doi.org/10.1016/0079-6611(88)90055-9, 1988.

Fouli, H. and Zhu, D.: Interfacial waves in two-layer exchange flows downslope of a bottom sill, Journal of Fluid Mechanics, 680, 194–224, https://doi.org/10.1017/jfm.2011.155, 2011.

Gacic, M., Rubino, A., Ursella, L., Kovacevic, V., Menna, M., Malacic, V., Bensi, M., Negretti, M., Cardin, V., Orlic, M., Sommeria, J., Viana Barreto, R., Viboud, S., Valran, T., Petelin, B., and Siena, G.: Impact of the dense water flow over the sloping bottom on the open-sea circulation: Laboratory experiments and the Ionian Sea (Mediterranean) example, Ocean Sci., 17, 975–96, https://doi.org/10.5194/os-17-975-2021, 2021.

Hilt, M., Auclair, F., Benshila, R., Bordois, L., Capet, X., Debreu, L., Dumas, F., Jullien, S., Lemarié, F., Marchesiello, P., Nguyen, C., and Roblou, L.: Numerical modelling of hydraulic control, solitary waves and primary instabilities in the Strait of Gibraltar, Ocean Modelling, 151, 101 642, https://doi.org/https://doi.org/10.1016/j.ocemod.2020.101642, 2020.

Lawrence, G. A.: The hydraulics of steady two-layer flow over a fixed obstacle, Journal of Fluid Mechanics, 254, 605–633, https://doi.org/10.1017/S0022112093002277, 1993.

Negretti, M. E., Zhu, D. Z., and Jirka, G. H.: Barotropically induced interfacial waves in two-layer exchange flows over a sill, Journal of Fluid Mechanics, 592, 135–154, https://doi.org/10.1017/S0022112007008324, 2007.

Negretti, M. E., Zhu, D., and Jirka, G.: The effect of bottom roughness in two-layer flows down a slope, Dyn. Oceans Atm., 45, 46–68, 2008.

Negretti, M. E., Flòr, J.-B., and Hopfinger, E. J.: Development of gravity currents on rapidly changing slopes, Journal of Fluid Mechanics, 833, 70–97, https://doi.org/10.1017/jfm.2017.696, 2017.

Pierini, S., De Ruggiero, P., Negretti, M., Sommeria, J., Schiller Weiss, I., Weiffenbach, J., and Dijkstra, H.: Laboratory experiments reveal self-sustained intrinsic oscillations in ocean relevant rotating fluid flows, Nature Sci. Rep., in press, 2022.

Pirro, A., Menna, M., Mauri, E., R., L., Salon, S., Bosse, A., Martellucci, R., Viboud, S., Valran, T., Hayes, D., Speich, S., Poulain, P., and Negretti, M.: Rossby waves driven by the Mid Mediterranean Jet impact the Eastern Mediterranean mesoscale dynamics., Nature Sci. Rep., 14, 29 598, https://doi.org/10.1038/s41598-024-80293-6, 2024.

Prastowo, T., Griffiths, R., Hughes, G., and Hogg, A.: Mixing in exchange flows through a contraction, 2006.

Pratt, L. J.: Hydraulic Control of Sill Flow with Bottom Friction, Journal of Physical Oceanography, 16, 1970 – 1980, https://doi.org/10.1175/1520-0485(1986)016<1970:HCOSFW>2.0.CO;2, 1986.

Pratt, L. J.: Critical conditions and composite Froude numbers for layered flow with transverse variations in velocity, Journal of Fluid Mechanics, 605, 281–291, https://doi.org/10.1017/S002211200800150X, 2008.

Pratt, L. J. and Helfrich, K.: Generalized Conditions for Hydraulic Criticality of Oceanic Overflows, Journal of Physical Oceanography, 35, 1782 – 1800, https://doi.org/10.1175/JPO2788.1, 2005.

Roustan, J.-B., Bordois, L., Dumas, F., Auclair, F., and Carton, X.: In Situ Observations of the Small-Scale Dynamics at Camarinal Sill—Strait of Gibraltar, Journal of Geophysical Research: Oceans, 128, e2023JC019 738, https://doi.org/https://doi.org/10.1029/2023JC019738, 2023.

Roustan, J.-B., Bordois, L., Garciá-Lafuente, J., Dumas, F., Auclair, F., and Carton, X.: Evidence of Reflected Internal Solitary Waves in the Strait of Gibraltar, Journal of Geophysical Research: Oceans, 129, e2023JC020 152, https://doi.org/https://doi.org/10.1029/2023JC020152, 2024a.

Roustan, J.-B., Bouruet-Aubertot, P., Bordois, L., Cuypers, Y., Carton, X., Dumas, F., and Auclair, F.: Turbulence Over Camarinal Sill and Its Impact on Water Mixing—Strait of Gibraltar, Journal of Geophysical Research: Oceans, 129, e2023JC020 709, https://doi.org/https://doi.org/10.1029/2023JC020709, 2024b.

Rubino, A., Gacic, M., Bensi, M., Kovacevic, V., Malacic, V., Menna, M., Negretti, M. E., Sommeria, J., Zanchettin, D., Barreto, R. V., Ursella, L., Cardin, V., Civitarese, G., Orli?, M., Petelin, B., and Siena, G.: Experimental evidence of long-term oceanic circulation reversals without wind influence in the North Ionian Sea., Sci. Rep., 10, 1905, https://doi.org/https://doi.org/10.1038/s41598-020-57862-6, 2020.

430 Sannino, G., Pratt, L., and Carillo, A.: Hydraulic Criticality of the Exchange Flow through the Strait of Gibraltar, Journal of Physical Oceanography, 39, 2779 – 2799, https://doi.org/10.1175/2009JPO4075.1, 2009.

Sannino, G., Sanchez Garrido, J. C., Liberti, L., and Pratt, L.: Exchange Flow through the Strait of Gibraltar as Simulated by a $\sigma$-Coordinate Hydrostatic Model and a z-Coordinate Non-hydrostatic Model, chap. 3, pp. 25–50, American Geophysical Union (AGU), ISBN 9781118847572, https://doi.org/https://doi.org/10.1002/9781118847572.ch3, 2014.

435 Shi, H., Negretti, M., Chauchat, J., Blanckaert, K., Lemmin, U., and Barry, D.: Unconfined Plunging Process of a Hyperpycnal River Flowing into a Lake: Laboratory Experiments and Numerical Modelling, Water Res. Research, 58, https://doi.org/10.1029/2022WR032633, 2022.

Sánchez-Garrido, J. C., Sannino, G., Liberti, L., García Lafuente, J., and Pratt, L.: Numerical modeling of three-dimensional stratified tidal flow over Camarinal Sill, Strait of Gibraltar, Journal of Geophysical Research: Oceans, 116, https://doi.org/10.1029/2011JC007093, 2011.

Tassigny, A., Bordois, L., Carton, X., and Negretti, M.: Entrainment and mixing generated at the Gibraltar Strait's exit. Part I: Camarinal Sill,
440 Deep Sea Research Part I.

Tassigny, A., Bordois, L., Carton, X., and Negretti, M.: Internal solitary waves in a realistic laboratory model of the Strait of Gibraltar, Dynamics of Oceans and Atmospheres, p. submitted, 2026.

Wesson, J. C. and Gregg, M.: Mixing at Camarinall Sill in the Strait of Gibraltar, Journal of Geophysical Research: Oceans, 99, 9847 – 9878, 1994.

445 Wirth, A.: On the hydrostatic approximation in rotating stratified flow, Nonlinear Processes in Geophysics, 32, 261–280, https://doi.org/10.5194/npg-32-261-2025, 2025.

Zhu, D. Z. and Lawrence, G. A.: Hydraulics of Exchange Flows, Journal of Hydraulic Engineering, 126, 921–928, https://doi.org/10.1061/(ASCE)0733-9429(2000)126:12(921), 2000.

---

## Author Comment (AC3)

**Responses to Reviewer 1**

*Review of "A realistic physical model of the Strait of Gibraltar" by A. Tassighy et al. A very interesting paper that presents the best laboratory physical model of the Strait of Gibraltar ever made so far. Congratulations to the authors. The experiment provides such an amount of experimental data that authors face a serious problem to select which ones are going to be pre-*
5 *sented in the manuscript, which one is the logical sequence to show the results, how are they connected to each other and so on. The final manuscript focus on tidal dynamics for different tidal forcing and compare the results with those already published in papers that deal with the same issue, either from observational or numerical approaches. There are a large number of experimental data related to high-frequency dynamics that are not addressed here (they will be covered in other ongoing articles) for obvious reasons of length. The analysis of the tidal regime alone has resulted in this extensive manuscript, which*
10 *will be read with great interest by the oceanographic community working in this very special environment. The manuscript should be published in OS, although authors must revise it with care. Below is a list of (mostly) minor or very minor points, whose extensive length is a consequence of the interest with which I have read the manuscript.*

We are really grateful to the referee for her/his thorough review and the positive comments. Her/his comments helped us to
15 make the manuscript more clear and refining certain discussions such as about the composite Froude number and the transports and fortnightly modulation of the baroclinic velocity.
Given the general comment of the referee about the extensive dataset of the HERCULES experiment and the difficulty to organize and present the results in a first, general, overall dynamics paper, which has to include extensive description of the physical model as well, we added a perspective paragraph in the conclusion section to describe better the different processes and areas
20 considered in the measurements, encouraging members of the community to contact us for using the available experimental data.
A point by point answer to each of the referee's raised issue is given below in blue text. All changes have been highlighted by blue text in the revised manuscript.

25 *Comments to the manuscript (following the order of writing)*

*L35. "the Mediterranean Outflow Water strengthens the Atlantic Meridional Overturning Circulation" I wouldn't dare be so categorical. There is still considerable debate about the role of the Mediterranean outflow in the AMOC.*
We replaced 'strengthens' with 'contributes to'.
30

*L46. "observationally-based" rather than "experimental"?*
We added 'based on observational data'.

*L78-86. A short phrase stating the origin, orientation and positive direction of x and y-axes would be welcome. Origin is*
35 *CS summit, as stated in line 216 later, but perhaps it is a good idea to mention it here. It seems rather obvious that orientation is along ("x") an across ("y") Strait, with positive directions eastward and northward, respectively. However, it is not until caption of Figure 4 (around L400) that this convention is explicitly said. The reason of this suggestion is that, here and there in the manuscript appear contradictions (writing errors?) regarding velocity and coordinate signs. Therefore, the convention must be clearly stated and this Section seems to be the right place.*
40 We added at the beginning of section 2 'The coordinate origin is set at the Camarinal Sill (CS) summit. A right-handed coordinate system is used, with the x-axis oriented eastward and the y-axis oriented northward.'

*Although not explicitly stated, the Atlantic and Mediterranean waters have the same temperature in all simulations, right? This makes density solely a function of salinity, which is important in certain parts of the work (i.e., transports). It would be*
45 *appropriate to mention this at some point in Section 2.*
We added at old line 270 'both at the same temperature of $\approx 18°$ so that the density is a function of salinity only in the experiment'.

*L122. "In the vertical momentum equation (3c)..." equation (3c) is horizontal momentum isn't it?* Done.

*L 135-136. "and Hs ≃ 100m is the half water depth at CS" Actually, water depth at CS is nearly 300m. Why this scaling factor of 100m? Does it have to see with the mean thickness of the Mediterranean layer at CS?*
For the scaling, the order of magnitude is important, so we choose 100m for simplicity without any loss of generality.

*L158. A "s" is missing. 35.77s* Done.

*L163. The Strouhal number is new to me. Other readers may encounter the same difficulty of understanding what does it mean. A definition and relevance of this number (similar to the one given for the Burger number in lines 151-153) would be welcome*
We added at line 143 when defining the Strouhal number
'$Ro = U/(fL)$, $Re = UH/\nu$ are the Rossby and Reynolds numbers, respectively, and $St = L/(UT_{tide})$ is the Strouhal number which compares the time scale of the oscillation (tide) and the advection time scale, and hence it is a measure of the unsteadiness of the flow.'

*L174. " .. on the slope angle.." Bottom slope? Interface slope?* Yes, bottom slope, we added it.

*L223. Delete "is"* Done.

*L229-230. References within brackets.* Done.

*L242. 6% is a clear overestimate. Tidal transport exceeds 3Sv (if you refer to the amplitude of the oscillation). 1.5% - 2% is more realistic.* We thank the referee for this precision, we replaced 6% with 1.5% - 2%

*Figure 1. Nice Figure showing the core of the experimental team (and the infrastructure). I like it.*
We thank the referee for this appreciation, we put this picture to give an idea of the size of the model.

*L249. Are units of the density difference ("kg/m3") correct? In fact, 0.19kg/m3 is a very tiny difference, ten times smaller than in the actual ocean, which contradicts the statement of L143-144 about enhancing the density difference (and, also, units in the color scale of some figures, i.e., Fig.9).*
The referee is right, the value is 19kg/m3 and not 0.19, we corrected this in the text.

*L260-261. No continuous red line in Figure 2a. Grey line perhaps, as stated in the caption? And y-axis indicate density difference, not salinity. On the other hand "..(25cm in the ocean).." is 25m, right?*
The referee is right, we corrected all this in the text and in the caption.

*L276. " ... of the order of magnitude of 1m/s..." Where? At CS? Please indicate.*
Yes, the value refers at CS, so we added 'at CS'.

*L279. What is it meant by "purely barotropic velocity" here?. Does it refer to the amplitude of a purely barotropic tide, i.e., the tide resulting if both Atlantic and Mediterranean basins are filed with homogeneous water (no baroclinic forcing at all)?*
The referee is right, we have made the precision in a footnote on page 12 .3as follows
'The purely barotropic velocity was measured in ad-hoc experiments in which the full rotating tank was filled with homogeneous fresh water with $\rho_0$ and only the barotropic forcing was applied.'

*Figure 2 caption. (L-3 says "salinity measurement" while y-axis indicate density difference. L7 mention SSH variations, L8 mention SSH anomaly. What is the difference between both SSHs?Apparently, they refer to the same process. L9-10 about the phase shift: see comment L305 below.*

We replaced salinity with density measurement. 'SSH variations' and 'SSH anomaly' refer to the same quantity so we replaced with 'SSH anomaly' everywhere in the text and in the caption.

*L291 No "red curves" in Figures 2c,d, but cyan or grey.* Done.

*L305 A phase shift of 0.2Ttide between the Atlantic basin and Tarifa station is declared. Taking Ttide the period of M2 (12.4h approx.) the shift would be 2.5h or 70° phase difference for M2 frequency. In the actual ocean, this phase differ-ence is hardly 15°, much less than the one reported in the manuscript (consult harmonic constants in the interactive web https://portus.puertos.es of Puertos del Estado, SPAIN). In the real ocean the phase difference is caused by the Earth rotation in order to maintain geostrophic balance at tidal scale, achieved via changing the sign of the cross-strait free surface slope at tidal time-scale. Moreover, as mentioned in lines 285-289, the relationship between the sea-surface oscillation and the tidal barotropic current in the actual ocean cannot be reproduced in the laboratory. In the experiment, eastwards flow corresponds to rising tide in the Atlantic, just the opposite that in the ocean. That means that the Coriolis force causing the cross-strait slope (and, hence, the phase shift in Tarifa) acts in opposite direction in the real ocean and in the experiment, which invalidates the comparison of results. I suggest removing these comments both here and in the caption of Figure 2.*
The referee is right: since we are not in similarity with respect to the external Froude number and because of the opposite sign between the SSH and the sign of the barotropic velocity, the comparison is not possible. So we removed the sentence in the text and in the figure's caption as proposed by the referee.

*L379. When speaking of mean density anomaly, do you refer to the ratio $\Delta\rho/\Delta\rho_0$ that appears in the top panels of Figure 4 (where $\Delta\rho = \bar{\rho} - \rho_0$ $\bar{\rho}$ being the averaged density over the seven tidal cycles ($\Delta\rho_0$ is already defined), right?). If so, it is not a density anomaly, but a ratio of density differences (dimensionless). Assigning a symbol to this anomaly, which is frequently discussed in the text, would make it easier to read.*
As suggested by the referee, we have defined the dimensionless density anomaly ratio $\rho^* \equiv \Delta\rho/\Delta\rho_0$ and added in the text '... highlights the mean dimensionless ratio of density differences $\rho^* \equiv \Delta\rho/\Delta\rho_0 = \bar{\rho} - \rho_0/(\rho_M - \rho_0)$, with $\bar{\rho}$ being the aver-aged density over the seven considered tidal cycles.' We have then replaced it where mentioned with the new defined variable $\rho^*$.

*L388. When saying " ..waters are more diluted (about 30%) etc..", do you mean that the density anomaly ratio is 0.3?*
Yes, but this corresponds to the dilution of the Mediterranean waters with respect to the initial value. We have precised in the text 'of about $0.3\rho^*$' and evrywhere when indicated in %.

*L390. "surface" instead of "layer"* Done.

*L391. The grey line, according to Figure 4 caption, is $\Delta\rho/\Delta\rho_0 = 0.5$ (dimensionless), which corresponds to $\Delta\rho = \Delta\rho_0/2$ (units of kg/m3). Please modify so that text and figure caption do not disagree. Moreover, this specific value is used (implicitly) to define the pycnocline, as stated later in lines 413-414. In my opinion, what is said in these two last lines should be moved there (to the end of line 391, more or less).* Done.

*L399. "deepen" or "descend" instead of "lower"?* Done.

*Figure 4 caption. L3 "... with the averaged velocities..." Actually, it shows the vector sum of averaged u and w with the later largely exaggerated. The same applies to Figure 8 and also to Figures B1 and B2 in appendix B.*
The quiver plot uses different scaling in the vertical and horizontal directions; consequently, one centimeter in the vertical does not correspond to the same graphical distance as one centimeter in the horizontal. This introduces an ambiguity in how the arrow angle can be defined. Two main conventions exist for specifying arrow orientation. In this study, we adopt the data-coordinate convention, so that a velocity vector tangent to the seabed is displayed as an arrow tangent to the bottom topography. Because the figures are intentionally stretched in the vertical to enhance readability, preserving geometric angles would pro-duce misleading impressions of the true flow direction. Arrow length is therefore determined solely by the in-plane speed, independent of any angle distortion. There is no unique or fully objective method for representing velocity vectors, and the

convention used here is relatively common. For clarity, we have now added this explanation to the manuscript and consistently refer to the quiver representations as "in-plane velocity vectors" throughout the text.

*L5 I guess you speak of the time-averaged composite Froude number, do you? Please indicate.* Yes, we specify now in the text that it is time averaged.

*L405-406 The mean thickness computation in equation (9) requires the definition of an interface depth, which seems to be the pycnocline defined as $\Delta\rho/\Delta\rho_0 = 0.5$. This is said later in lines 413-414, but should be said before. Moving these lines (as I suggested above) will improve the text understanding.*
We thank the referee for this comment (and those above related). We rewrote this paragraph for more clarity (also in consideration of the above referee's comments) and added in the appendix E the different computations of the Froude number. Since the most reliable results that are also in accord with the base assumption for the theoretical derivation of the composite Froude number, we replaced the old figures with the Froude number, that were computed with the constant initial density difference, using the local density difference instead. Please see appendix E for further details.

*L423-424. "... given in figures B1 and B2 in the appendix". Remove ", respectively, that are given"* Done.

*L437. "horizontal velocity" rather than "horizontal flow". I have seen that the term "horizontal velocity" usually refers to the rotated (along-strait) component of the horizontal velocity throughout the text; particularly in the second half of the paper this seems to be the rule. But not always means this component. For instance, it does not in the aerial view maps of Figures 5 and 7, which actually show the "horizontal velocity". Perhaps it is convenient to state a criterion somewhere in section 2 or keep the notation "u-component" for the horizontal velocity in the second part (after presenting Figure 7). As you can see below in my review, there are a few comments addressing this point.*
In the presentation of the results from vertical sections, the horizontal velocity always refers to along-strait horizontal velocity as we are only measuring in-plane velocities. To avoid confusion with the horizontal planes presenting horizontal velocities, we replaced the 'horizontal' velocity spelling of the measurements in the vertical sections with 'along-strait' velocity.

*Figure 5 caption. L1 "Averaged horizontal velocity" instead of "Averaged meridional and zonal velocities". Insert "relative" before "vorticity"* Done.

*Figure 6 caption. Are the presented plots time-averaged values? Please indicate. L3 "dashed" should be "dotted" and "dotted" should be "dashed". The sentence "vertically integrated volume transport and below salt transport;" has nothing to see with the panels. Please, remove.* Done.

*L451-452. Remove the sentence "The bottom panels .... three transects" or, better yet, incorporate it at the caption of Figure 6.* We removed the sentence.

*L454. Remove "where ...... waters". Unnecessary; it has been already said.* Done.

*L466. "... to the shallowing pycnocline.." instead of "... to the decreasing pycnocline depth..."* Done.

*L472-473. "... 50% higher during neap tide compared to the spring tide condition with exception of the southern transect where values remain similar for all tidal conditions" This is not what can be deduced from Figure 6. Actually, it is in the southern transect where G2 differences between spring and neap tides are the greatest. Maybe you are referring to the northern transect in the caption?*
The referee is right. We replaced 'approximately' with 'up to' 50% and replaced 'southern' with 'northern' transect.

*L474. "x=-30cm and x=-45cm" instead of "x=30cm and x=45cm"? (this has to see with my comment to lines 78-86 above).*
195 *Nice to indicate the corresponding distances in km.* Done.

*L480. "horizontal velocity" instead of "zonal and meridional velocities"* Done.

*L480. "tidal average"? Such an average would provide a unique value for the averaged period, that is, for a tidal cycle,*
200 *whereas Figure 7 presents snapshots at different moments of a tidal cycle. Maybe authors are referring to the average of the seven resolved tidal cycles for "Outflow", "HWS", etc.?*
Yes, we are referring to the average over the four (for these horizontal measurements) resolved tidal cycles at the given phase. We have now specified it in the text and caption.

205 *L486. "...(latter two row panels)" Do you mean "third row", which is where inflow is presented*
We refer to both panels, so we added in the text 'during outflow and high water slack' and 'during inflow and low water slack'.

*L492. "averaged over the phases of maximum inflow" Again, do you refer to the maximum inflow averaged over the seven resolved cycles?*
210 Yes, we are referring to the average over the seven resolved tidal cycles at the given phase. We have now specified it in the text and caption.

*L497. "demonstrates that tidal forcing is capable of mixing the entire water column down to 500m". Not clear to me. I think you are referring to the periodic occupancy of the whole water column by Mediterranean waters (deep red colors in Figure 9).*
215 *It could be mixing, but, in my opinion, is the uprising of the interface during the outflow which can eventually reach the surface during spring tides. If so, it is advection rather than mixing. A different issue is that, in HERCULES experiment in particular, the whole Mediterranean layer is well mixed from the bottom to the interface (which eventually can reach the free surface) with clear density differences between inflow (light red) and outflow (deep red) periods. These differences can be of the order of the mentioned percentage (15-20%, normalizing the density differences by $\Delta \rho_0$ I guess. Please, indicate)*
220 We where not precise in the text. We wanted to show that tides are capable of advecting the full (mixed) water column to the east during inflow, causing periodic oscillations in the density of the order of $(0.15 - 0.2)\rho^*$. This is true also for the observational data, as seen from the colorbar, even if not homogeneously from cycle to cycle and down to the bottom. This is probably also due to the different composition of waters in the real ocean. We have rewritten the paragraph as follows:
'A closer inspection, presented in figure 9, demonstrates that tidal forcing is capable of advecting the full water column down
225 to 500m of mixed waters east of CS during spring tide, with oscillations of the density ratio as a function of the tidal phase of the order of $(0.15 - 0.2)\rho^*$. Since in the experiment we do not have different density components in the Mediterranean and in the Atlantic waters, in the experiment the whole Mediterranean layer is well mixed from the bottom to the interface with clearer differences between inflow and outflow.'

230 *L504. "water can still overflow" instead of "flow can still surmount"* Done.

*Figure 8 caption. Please, indicate what the bottom panel is.*
We added: 'The bottom panel indicates the variation of the depth integrated barotropic velocity at CS over a tidal cycle with the dots indicating the maximum inflow and outflow corresponding to the above panels for neap and spring tides.'
235

*Figure 9. Please, explain what are the numbers given at the left and right sides of the color scale. I guess, numbers on the left are the $\Delta \rho$ difference in the actual ocean, and numbers on the right are the same for waters in HERCULES experiment. If so, note that the values of the order of 10 kg/m3 that are representative of this scale are clearly contradicting (two order of magnitude greater) the values provided in line 249 for $\Delta \rho_0$ See my comment on L249 above as well.*
240 We added explanation about the values of the colorbar in the plot, that are right here. The initial density difference given at the beginning of 0.19 was wrong ans it has been corrected following the referee's comment above.

*L519. "The second-column panels show a clear offset between the pycnocline and the region of maximum velocity shear"*
*The pycnocline ($\Delta\rho/\Delta\rho_0 = 0.5$) is clearly seen in Figure 9. The region (surface?) of maximum shear is not so clearly depicted.*
*It can be inferred by the color contrast (reddish versus bluish shading, probably), but locating it with a certain degree of accuracy is not an easy visual issue. West of Camarinal this color contrast (shear "interface") is located in a well differentiated depth range than the pycnocline. However, both surfaces coincide acceptably well elsewhere. Maybe the comment is referring to the west of Camarinal specifically? Please clarify the quoted sentence above.*
The referee is right saying that it is difficult to see the surface of maximum shear just looking at the color contrast of the along-strait velocity. A more direct comparison is given later in Figure 11. We slightly modified the text saying that the colormap suggests this offset, but we refer then to Figure 11 where it can be clearly seen. And yes, the offset is present west of CS and up to 20cm east, so we precise this as well in the revised text. We have also added and additional figure in the appendix D with correlation maps between $S$ and $N$ at different tidal phases along-strait and for neap and spring tides in figure D1.

*L521-523. I have read the sentence "Vertical velocities ..... spring tide conditions" several times and I cannot catch its meaning and the reason to be here. Can you be a bit more explicit?*
The hydraulic jump west of the Camarinal Sill is characterized by an abrupt reversal in vertical velocity, transitioning from strongly negative to strongly positive values over a short distance. Consequently, the vertical velocity field provides a clear indicator of the location and intensity of the internal hydraulic jump. Our observations show that during neap tides, the vertical velocity attains higher maximum and minimum values within a confined region west of the sill. In contrast, during spring tides, the vertical velocity extremes are lower in magnitude but spread over a broader area. This behavior indicates that the hydraulic jump is displaced farther west during spring tides, whereas it remains confined to a more restricted region during neap tides. We have rephrased this sentence as follows
'The hydraulic jump west of CS is characterized by an abrupt reversal in vertical velocity, transitioning from strongly negative to strongly positive values over a short distance. Consequently, the vertical velocity field provides a clear indicator of the location and intensity of the internal hydraulic jump. The internal hydraulic jump is evident at the sill for both neap tide and spring tide maximum outflow conditions, along with two additional control points farther downstream (to the west), more pronounced during spring tide. Our observations show that during neap tides, the vertical velocity attains higher maximum and minimum values within a confined region west of the sill. In contrast, during spring tides, the vertical velocity extremes are lower in magnitude but spread over a broader area. This behavior indicates that the hydraulic jump is displaced farther west during spring tides, whereas it remains confined to a more restricted region during neap tides.'.

*L525. "even negative velocities"? Do you mean "even positive"? (cf, column 2, row4 in Fig 8, better seen in column 2, row4 of Figure 10)* Yes, correction added.

*L547-549. Same comments as in L419 above. I think the mismatch is limited to the west flank of CS.*
The mismatch is found during both spring and neap tides, west of CS and up to 20 cm (5Km) east of CS (see also appendix D where we add an additional figure displaying both N and S).

*L548. "...under spring tide". It rather seems "neap tide"*
In fact the decorrelation is always present but with different amplitudes for neap and spring tides and HWS and LWS (see also reply to the following comment). We rewrote the sentence as: '... for both spring tide and neap tide conditions, even if less marked in this latter case (see also figure 11 and C1). The same holds true during low-water slack.'

*Figure 11 caption. Are vertical profiles just above CS? Please indicate* Yes, we have made the precision in the text. We have also added an additional figure in the appendix D to show the correlation map between S and N at different tidal phases, to show that it is more marked for spring tide conditions and also present west and up to $x = 20cm$ east of CS.

*L572-573. "...or from velocity interface" Are you meaning "velocity interface (that is, u=0) or surface of maximum horizontal (u-component) velocity shear? The surface u=0 can eventually disappear during short periods of the tidal cycle and is useless for estimating layer thicknesses*

We write clearly now in the text that we refer to the 'maximum vertical gradient of along-strait velocity'.

*Figure 12 caption. "velocity" instead of "speed"* Done.

*L593. "(positive values of ub)"? isn't it "negative"?* Yes, correction made.

*L656. Add "(see the locations indicated in Figure 1)" after "conditions".* Done.

*Figure 13. As far as HERCULES experiment is unable to reproduce correctly the sea level phase relationship of the surface oscillation in the Atlantic and Mediterranean basins (lines 285-290 of the manuscript), it does not seem like a good idea to show the oscillation in both basins. I think the one in the Atlantic side is enough to delimitate the inflow and outflow periods in the Hovmöller diagrams below. I suggest to remove the Mediterranean sea level curve and add vertical dashed lines across the Hovmöller diagrams showing inflow and outflow periods.*

We agree with the referee, and we propose to display the depth integrated barotropic velocity $u_b$ at CS as done in the other plots. In figure 13, we remove the arrows showing the barotropic flow in the Hovmöller diagrams of the pycnocline as asked further below and make the figure more readable. This change between SSH and $u_b$ has been applied also to figure 14. Note that the figure 14 has additionally changed since we compute now the composite Froude number using a local $g'$ as explained in appendix E of the revised manuscript. Finally, In figure 15 we removed the SSH panels and we just keep the barotropic flow arrows in the Hovmöller diagrams of the buoyancy transport.

*Figure 13, caption. If possible, mark with dots the locations of M02 and M05 in the bathymetry left panel of the center section.*

Since we mention MO2 and MO5 in the discussion of figure 13 it makes sense to see the position of MO2 and MO5 in the vertical transect, so we add the position in the figure 13 as suggested. We think it is also appropriate to add it in figure 3a when defining the measurement regions.

*Figure 13, caption. "vertically averaged horizontal velocity" or "vertically averaged u-component of horizontal velocity"?*
In the presentation of the results from vertical sections, the horizontal velocity always refers to along-strait horizontal velocity as we are only measuring in-plane velocities. To avoid confusion with the horizontal planes presenting horizontal velocities, we replaced the 'horizontal' velocity spelling of the measurements in the vertical sections with 'along-strait' velocity.

*L660-661. "Moreover, we also observe that the position of this front does not happen at the same time in the three transects" I cannot see it in Figure 13. Perhaps some indications/marks in the very Figure would clarify the issue.*
*L662. I cannot identify the 2s shift mentioned in the text. Perhaps the marks I suggest above may help. . .*
As explained above, we removed the SSH panels and replaced them in the top panels with the barotropic flow. Consequently, the arrows on the Hovm"oller diagrams were also removed. Thanks to the referee's comment, we attempted a more quantitative estimate of the phase shift; however, we found it difficult to discern, and moreover, it varied from one tidal cycle to another. Therefore, as we are no longer confident in this result, we have removed it from the discussion throughout the manuscript and have updated the figure captions and corresponding text accordingly. We are investigating closer this issue in a forthcoming paper focusing on ISW. we have added instead in the revised manuscript line 712:
"We attempted a quantitative estimate of the phase shift using our experimental data; however, it proved difficult to identify and was found to vary from one tidal cycle to another, preventing any robust conclusion. This issue is more appropriately addressed in a forthcoming paper focusing on internal solitary waves (Tassigny et al., 2026)."

*L670. "Garcia-Lafuente et al. (1990, 2018); Sanchez-Roman et al. (2018); Roustan et al. (2024a)" within brackets.* Done.

*L673. ". . . with respect to the reported phase shift between north and south" Are you speaking of the sea-level phase differ-ence (i.e. barotropic tide)?*
We referred to the internal tide, as said just above. We cannot say anything about the external tide and the SSH shift since we

340 are not in similarity with the external Froude number. As said above, we removed this discussion in the revised manuscript.

*L680. "x=-30cm and x=-45cm"?* Yes. We corrected it in the text.

*L691. "x=-50cm"?* Yes. We corrected it in the text.

345

*L698. "...or the velocity interface, creating..." or " ...or the surface of maximum shear of the u-component of the horizontal velocity, creating.."* Done.

*L708. "the transport of salt." This is true as far as T=cte. I guess T=cte has been implicitly assumed in the study. I have*
350 *already made a comment on the interest of stating this constancy in Section 2. Nevertheless, it wouldn't hurt to mention it again here.*
The temperature within the tank is uniform. Density gradients are generated exclusively through variations in salt concentration, with ethanol added in some experiments to correct the refractive index. The Laser-Induced Fluorescence (LIF) technique allows measurement of Rhodamine concentration, which is directly related to fluid density following accurate calibration, as
355 described in Appendix A. In the ocean, as the referee notes, density gradients result from both salinity and temperature variations; therefore, the quantity $\Delta\rho, u$ does not correspond exclusively to salinity transport. Assuming linearity of the equation of state, $\rho = \rho_0 + \alpha(T - T_0) + \beta(S - S_0)$, it is in principle possible to decompose $\Delta\rho, u$ into a temperature transport contribution $\alpha, \Delta T, u$ and a salinity transport contribution $\beta, \Delta S, u$. To avoid any misunderstanding and unnecessary additional discussion, we have relabeled the "salt transport" as "buoyancy transport."

360

*L709, eq.(16). Limits of integrals in the equation are z=0 and z=bottom, aren't they? To mention the limits in the text (or in the equation) would be OK to avoid any confusion with transports in either layer.* Done.

*Figure 15 caption. "Mean transport contributions... " Add "(per width unit)" so that the sentence reads "Mean transport*
365 *(per width unit) contributions...." On the other hand, does "mean" here refer to tidally averaged? Also write "transects" instead of "planes"*
Mean refer to tidally averaged, as indicated by the operator defined in equation 8. Corrections made.

*Figure 16 caption. Indicate what the black arrows are. (Transports per width unit, aren't they?)*
370 Black arrows are the vertically averaged along-strait tidal velocities $u_b$, giving the reference of the tidal phase. We have added this sentence in the caption of Figure 16.

*L726. "..positive"? The mean (tidally-averaged?) transports in Figure 15 are mainly negative in all sections (north section could be a small exception). Figure 16 shows that the transport can be positive in spring tides in the eastern part, but only for*
375 *short periods. But, when averaging over a tidal cycle, it appears to be negative, as seen in Figure 15. So what does "positive" refer to in this line?*
We are sorry for this error, we should have written "negative" instead of "positive". Correction made.

*Lines 743-796. This part of the text should be a new sub-section 4.2.6 named "Time-variable tidal forcing amplitude" or*
380 *"Transient tidal forcing" or something similar. These lines clearly deal with a different topic than subsection "4.2.5 Transports"*
We originally addressed this point in the transport section because the issue arose from examining the studies of Vargas et al. (2004, 2006), who demonstrated that the amplitude of the net baroclinic flow varies with tidal amplitude and subsequently quantified associated transports. The referee is correct that, as presented, this discussion is not directly related to transport
385 estimates. We have therefore reorganized the manuscript by creating a new subsection, now Section 4.3, entitled "Fortnightly modulation of the baroclinic flow." The revised title with respect to the one proposed by the referee reflects our intent to include transient experiments only for reasons of direct comparison with observational data. In this paper, we deliberately avoid detailed analysis of transient dynamics (promising so far very interesting results), which will be the subject of a separate

manuscript.

*L751. "horizontal velocity component" I think "along-strait" or "u-component" is meant (line 755). Please, correct. Same applies to Figure 17 caption.* See response above about the same issue.

*L761. "...three moments..." instead of "...three points..."* We wrote 'three instants'.

*L763. "In the third row..."* Done.

*Figure 17 caption. Sentence "with a cutoff period....... Vargas et al., 2006)" is unnecessary as it is already in the text. I suggest a short "(see text)" instead. Add " in the transient tide experiment" after "(cyan line)" in line 7.* Done.

*L766. "... consistent with the reduced vertical velocities"? No vertical velocities are presented. Is it meant "...consistent with the reduced u-component velocity in the vertical profiles"?*
We removed the sentence, since it refers to an old figure in which we also showed the vertical velocities.

*L771. A similar comment about the sentence "...shape of the vertical velocity profiles"*
We replace "vertical velocity profiles" by " vertical profiles of along-strait velocity"

*L773-796. This part of the manuscript must be revised carefully. There is mention to Figures/panels that are not seen in the manuscript (i.e. L778 mention a "color scale" in Figure 18, as if this Figure were a sort of Hovmöller diagram);*
Figure 18 is the correct figure. The three tidal forcing are labeled with three different colors, the lighter corresponding with no tide and the darker to spring tide. We replaced "color scale" by "colored lines" and "shades" with "colors" for more clarity.
*line 773 indicates "Three possible mechanisms", when the text only addresses two;*
We did not mention the third one explicitly in fact, so now we do it in the revised manuscript.
*sentence in L786-789 "Consequently, the observed decrease in net baroclinic flux (flow?) at the sill cannot be attributed to tidal-to-mean energy transfer, as our results demonstrate the opposite effect" Apparently contradicts the previous sentence in L785 "...that the tidal flow transfers energy to the mean flow, thereby accelerating it".*
In fact, the last part of the sentence 'Consequently, the observed decrease in net baroclinic flow at the sill cannot be attributed to tidal-to-man energy transfer, as our results demonstrate the opposite effect' was misleading. We now replace it with 'Consequently, the observed decrease in net along-strait baroclinic velocity at the sill cannot be attributed to the tidal-to-mean energy transfer.'.
*The sentence "the flow tends to bypass the obstacle laterally rather than pass over it" in L790-791, when speaking, of a sill is confusing. How can a flow bypass laterally a sill? In my opinion, the flow either overflows the sill or remains stagnant in the upstream side*
When the tidal strength increases, there is a larger flow fraction passing at the shallower sides of the sill (which is of course 3D), than that which overflows the highest part of the sill. In order words, the transports in the southern and northern transects increase for increasingly high tidal amplitude as compared to the middle transect which passes through the summit of the sill, as it is also shown in figure 15. We have now replaced the text as 'more water circumventing the sill than overflowing it above the summit'.
*And an important question to be clarified, does the tidal flow in this final part have to see with the eddy-fluxes already discussed in several published articles in the literature. It seems that yes, but authors should clarify or comment the issue.*
We thank the referee for this relevant remark. Previous studies state that an increase in tidal amplitude leads to a decrease in the net buoyancy transport, which is instead compensated by tidal transport via eddy fluxes. In the present study, we interpret the concept of eddy fluxes as representing the tidal and turbulent transports defined in equation (16) and shown in figure 16 (third and fourth rows, respectively).
As discussed in the text, the tidal and turbulent transports are approximately one order of magnitude smaller than the volume and buoyancy transports. In particular, the tidal transport is of order $\approx 5 \cdot 10^{-3}\,\mathrm{kg\,m^{-3}}$, while the turbulent transport is even smaller, whereas the buoyancy transport reaches values of order $\approx 5 \cdot 10^{-2}\,\mathrm{kg\,m^{-3}}$. Although we observe an increase in tidal

transport with increasing tidal strength, this increase is not sufficient to explain the corresponding reduction in buoyancy transport.

440 Therefore, the diminished net baroclinic velocity reported in our experiments, as well as in observational data when tidal forcing is applied, cannot be attributed to the action of eddy fluxes alone. This discussion has now been added to the revised manuscript and is proposed as a motivation to investigate additional mechanisms that may contribute to the reduced net baroclinic velocity. We added this in the discussion presented in the new section 4.3:

'A third mechanisms which may explain the reduced net baroclinic velocity when increasing the tidal amplitude are tidally
445 driven *eddy fluxes*, which can account for large portions (up to 40–60%) of the total exchange transport (Bryden et al., 1994; Vargas et al., 2006) in regions where hydraulic control is intermittently lost, such as at Camarinal. These studies state that an increase in tidal amplitude leads to a decrease in the net buoyancy transport, which is instead compensated by tidal transport via the *eddy fluxes*. Tsimplis and Bryden (2000) and Bryden et al. (1994) documented strong tidal variability and interface depth oscillations at CS, which are associated with significant fortnightly and monthly fluctuations in layer transports that reflect
450 the influence of tidal eddying on exchange dynamics. These *eddy fluxes*, arising from the positive correlation between vertical interface and tidal current fluctuations, contribute to augment the net transport beyond what is predicted by traditional, steady two-layer hydraulic theory.

In the present study, we interpret the concept of *eddy fluxes* as representing the tidal and turbulent transports defined in equation (15) and shown in figure 15 (third and fourth rows, respectively).
455 As discussed above with figure 15, the tidal and turbulent transports are approximately one order of magnitude smaller than the volume and buoyancy transports. In particular, the tidal transport is of order $\approx 5 \cdot 10^{-3}\,\mathrm{kg\,m^{-3}}$ (the turbulent transport is even smaller), whereas the buoyancy transport reaches values of order $\approx 5 \cdot 10^{-2}\,\mathrm{kg\,m^{-3}}$. Although we observe an increase in tidal transport with increasing tidal strength (cf. third row of figure 15), this increase is not sufficient to explain the corresponding reduction in buoyancy transport.

460 Therefore, the diminished net baroclinic velocity reported in our experiments, as well as in observational data when tidal forcing is applied, cannot be attributed to the action of *eddy fluxes* alone.

*I have no posted comments to "Conclusions" section, as it should be partially rewritten by the authors in the light of the list of comments above. Neither have I revised Appendix A, as it addresses issues far from my expertise.*
465 We slightly modified the conclusion section according to the referee's comments and added an overview of the measured areas and processes for future work. We merged the three figures of appendix A in one unique figure.

*L904, Appendix B. ". . . indicate the in-plane velocity vectors" Actually it shows the vector sum of u and w components (with the later largely exaggerated) in the northern transect. The same applies to Figure B1 caption for the southern transect.*
470 See the response above relative to figure 4.

**References**

Tassigny, A., Bordois, L., Carton, X., and Negretti, M.: Internal solitary waves in a realistic laboratory model of the Strait of Gibraltar, Dynamics of Oceans and Atmospheres, p. submitted, 2026.

---

## Author Comment (AC4)

**To the Editor**

We are very grateful to all three referees for the careful review and the positive comments. A point by point answer to each of the referees raised issues is given in blue text in the response documents. All changes have been highlighted by blue text in the revised manuscript.